PERSPECTIVE

# Nanohybrids of atomically precise metal nanoclusters

Koustav Sahoo[1], Tapu Raihan Gazi [1], Soumyadip Roy[1] &
Indranath Chakraborty [1✉]

Atomically precise metal nanoclusters (NCs) with molecule-like structures are emerging nanomaterials with fascinating chemical and physical properties. Photoluminescence (PL), catalysis, sensing, etc., are some of the most intriguing and promising properties of NCs, making the metal NCs potentially beneficial in different applications. However, long-term instability under ambient conditions is often considered the primary barrier to translational research in the relevant application fields. Creating nanohybrids between such atomically precise NCs and other stable nanomaterials (0, 1, 2, or 3D) can help expand their applicability. Many such recently reported nanohybrids have gained promising attention as a new class of materials in the application field, exhibiting better stability and exciting properties of interest. This perspective highlights such nanohybrids and briefly explains their exciting properties. These hybrids are categorized based on the interactions between the NCs and other materials, such as metal-ligand covalent interactions, hydrogen-bonding, host-guest, hydrophobic, and electrostatic interactions during the formation of nanohybrids. This perspective will also capture some of the new possibilities with such nanohybrids.

The chemistry of atomically precise metal nanoclusters (NCs), especially gold, silver, and copper NCs become a promising research field over the past few years because of their unique physical, chemical, and optical properties[1–7]. These NCs consist of a few to hundreds of metal atoms. They are composed of a metal core linked by staple motifs, further anchored by ligands such as thiolates, phosphines, carbenes, alkynyl, or a mixture of ligands[1,8–11]. Their stability can be explained by the super atom theory[12,13]. NCs can have specific chemical formulas as $[M_xL_y]^z$, where x, y, and z refer to the number of metals (M), ligands (L), and charges on the NC[1]. Mass spectrometry and single-crystal X-ray diffraction are the best methods to confirm their atomically precise nature. Recent mass spectrometry advancements can also help characterize poly-disperse NCs irrespective of their atomicity[1,14]. Because of their tiny size (<2 nm), comparable to the size of fermi electrons, the continuous energy levels break into discrete energy levels, creating a connection between the nanoparticles and the atoms[15]. Thus, they show interesting optical and electronic properties, including photoluminescence(PL)[16–22], catalytic activity[23–28], and magnetism[29–33]. The photoluminescence quantum yields (PLQY) of metal nanoclusters are relatively low in comparison to corresponding semiconductor quantum dots (QDs), and their emission lifetime is typically in the nanoseconds (ns) regime[22]. However, these materials exhibit good photo-stability, bio-compatibility, facile synthesis techniques, large stokes shifts, and tunable fluorescence intensity[6,34–41]. There are now many reported methods to improve their PLQY, especially via doping, surface rigidification[42], or ligand engineering[16] which makes them useful for many PL-based applications[1,43]. They have wide applications in various fields such as chemical sensing, cell

[1] School of Nano Science and Technology, Indian Institute of Technology Kharagpur, Kharagpur 721302, India. ✉email: indranath@iitkgp.ac.in

labeling, bio-imaging, phototherapy, drug delivery, catalysis[17,20,44–50], photovoltaics, and single electron spectroscopic applications[21,22,51–53].

Among the NCs, gold nanoclusters (AuNCs) have been the subject of most research (>50% of the published data) because of their unique optical and catalytic features and relatively lower reactivity[54–62]. Schaaff et al. first reported atomically precise glutathione-protected (SG) AuNCs in 1998 and assigned the cluster as $Au_{28}(SG)_{16}$. This discovery gives a new direction in this atomically precise NCs research field[63]. Numerous numbers of NCs are now discovered with their detailed mass spectrum, and many of them have been characterized by single x-ray crystallography, such as $Au_{18}(SR)_{14}$, $Au_{20}(SR)_{16}$, $Au_{23}(SR)_{16}$, $Au_{24}(SR)_{16}$, $Au_{25}(SR)_{18}$, $Au_{28}(SR)_{20}$, $Au_{30}(SR)_{18}$, etc[1,3].

Similarly, in the case of silver nanoclusters (Ag NCs), the optical features, intense luminescence, and antibacterial activity have received a lot of attention[64–68]. Recently, copper nanoclusters (CuNCs) are emerging due to their comparable optical properties and low cost[69,70]. However, unsaturation in coordination and electronic inaccessibility makes the NCs more chemically reactive. Therefore they are thermodynamically unstable[71]. Conversely, poor electron-hole pair recombination efficiency results in low fluorescence intensity and quantum yield (QY) compared to semiconductor quantum dots[72–75]. NCs also suffer from long-term stability, often required for practical application, especially in devices. Therefore, hybrid nanomaterials containing NCs have recently gained attention as a promising new class of multifunctional materials that can be used in many applications[76–79].

In general, hybrid nanomaterials are made up of two or more organic or inorganic components that are typically linked at the nanoscale level by covalent bonds or noncovalent bonds (electrostatic interactions, hydrogen-bonding, van der waals interactions, and so on)[80]. Integrating diverse materials within a single material yields hybrid materials with multifunctional and distinctive characteristics. This is attributed to the emergence of new or synergistic/anti-synergistic effects in their properties, which are not present in the individual components. Different strategies are reported to synthesize NCs-based nanohybrids, which can be categorized based on the type of interactions between the NCs and other nanomaterials, e.g., class-I nanohybrids (covalent interactions), class-II nanohybrids (noncovalent interactions such as hydrogen-bonding interactions, electrostatic interactions), and class-III nanohybrids (other types of interactions such as hydrophobic interactions, host-guest interactions, etc.) (Fig. 1)[81–96]. Such nanohybrids can exhibit exciting physical and chemical properties such as high photosensitivity[97], porosity[98], chemical stability[99], PL property[100], etc. As a result, these nanohybrids became a hot topic, especially in applied research such as catalysis, sensing, light harvesting, and so on[27,101–111]. So far, only a few reviews exist on nanocluster-based materials (especially composites), mostly on the synthesis and application aspects[76–79]. This perspective attempts to categorize the nanohybrids based on the type of interactions between the atomically precise metal NCs and other nanomaterials, together with some exciting properties and applications of the nanohybrids.

**Class-I nanohybrids (hybridization via metal-ligand covalent interactions).** NCs are protected by ligands, the first point of contact for any hybridization process with other nanomaterials. These ligands play a crucial role in the resultant geometry and properties of the nanohybrids depending on the type of interactions (chemical or physical). The interactions between ligands and other nanomaterials in the nanohybrids can be categorized into covalent interactions and non-covalent interactions (hydrogen-bonding, electrostatic interactions, pi–pi interactions etc.) and other interactions (host–guest interactions, hydrophobic interactions). Som et al[81]. reported the synthesis of nanohybrids by the interaction of atomically precise $Na_4Ag_{44}(p-MBA)_{30}$ NCs (where p-MBA is para-mercaptobenzoic acid) with Te nanowires (NWs). In their report, the carboxyl group of the p-MBA was found to be anchored on the TeNW surfaces via the Te-O covalent bonds, which were the point of contact for the nanohybrid formation. The resulting nanohybrids displayed optical properties of both the NCs and NWs. In this case, not all thirty ligands of the $Ag_{44}$ NCs have been involved in the ligand (NCs)-metal (Te) covalent interaction; some remained accessible. The carboxylic groups of these accessible p-MBA ligands were further used in self-assembly in a liquid-air interface via hydrogen-bonds (H-bonds) formation. The H-bond forms between the nearby $Ag_{44}$ NCs on the same or different Te NWs. The extent of coverage of NCs on the NWs surface and these H-bonds determine the final structure of the self-assembled nanohybrid. The TEM image suggested the formation of cross-bilayer structures (Fig. 2a) where the NWs from the same layer lie parallel but are at an unusual angle of ≈81° to NWs from the second layer, and the inter-NW distance was about 3.4 ± 0.3 nm. The formation of such sandwich-like structures of clusters offers a more orderly arrangement, which provides the hybrid with stability for over a few weeks. These self-assembled nanohybrids can be used as strain sensors when these thin films can be transferred to a polyethylene terephthalate (PET) substrate.

Rodrigues et al[82]. integrated the red luminescent atomically precise AgNCs ($Ag_{29}(DHLA)_{12}$) (DHLA: dihydrolipoic acid) on the surface of a tetrapod (Tp)-shaped three-dimensional (3D) ZnO particle surfaces (ZnO Tp) and investigated the change in structural and luminescence properties of the formed hybrid materials (Fig. 2b). In the nanohybrid, the $Ag_{29}(DHLA)_{12}$ NCs are anchored homogeneously to the ZnO surface via covalent

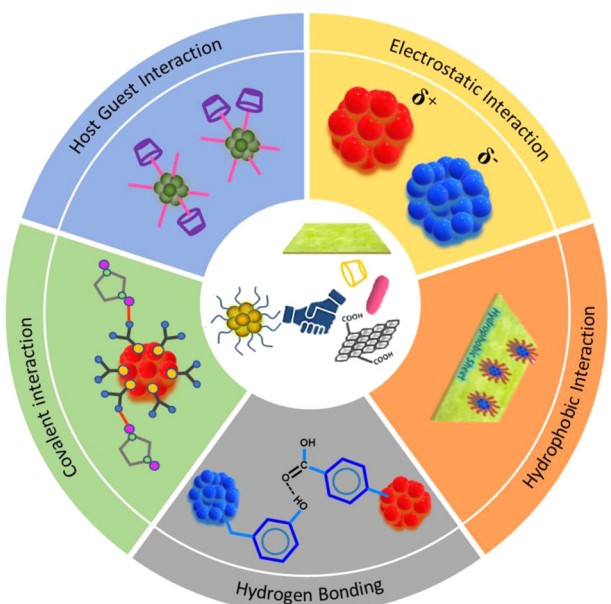

**Fig. 1 Schematic representation of nanohybrids with different kinds of interactions existing between NCs and nanomaterials.** A general scheme of the formation of nanohybrids of atomically precise nanoclusters is shown at the center and all the segments describe the different interactions such as electrostatic interaction, hydrophobic interaction, hydrogen bonding, covalent interaction, and host–guest interaction that exist between the nanoclusters and other nanomaterials during the formation of the nanoclusters.

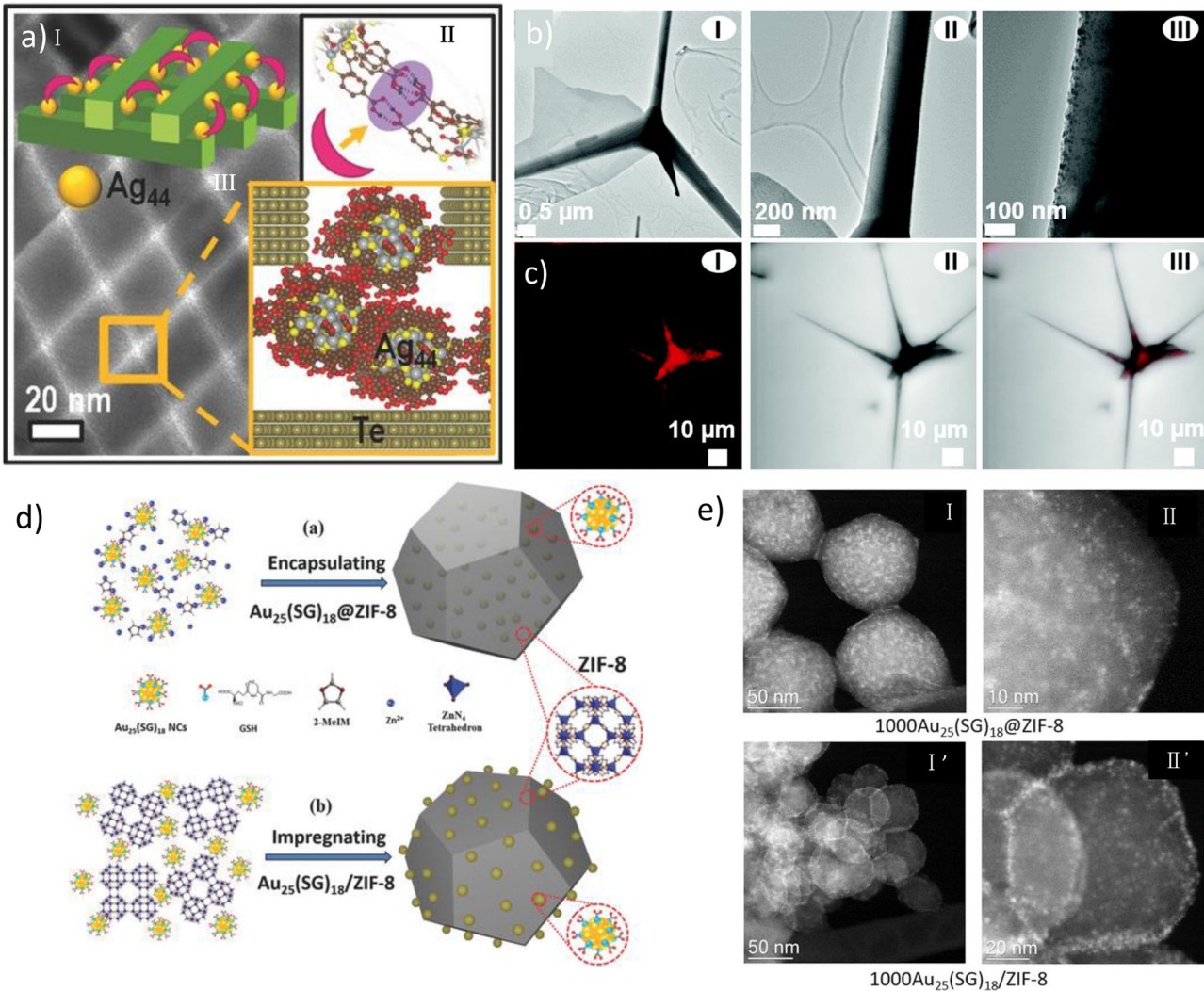

**Fig. 2 Nanohybrids based on covalent interactions. a** TEM image of the crossed assembly formed with Ag$_{44}$@Te NWs for the same NW concentration as the pristine Te NWs. and (I) a sandwich structure of the nanohybrid, (II) H-bonding between neighboring clusters in the Ag$_{44}$(p-MBA)$_{30}$ superlattice structure and (III) Schematic representation of the 81° orthogonal bilayer assembly, respectively. Reprinted with permission from ref. [81], Copyright 2016 WILEY. **b** TEM images of the AgNC@ZnO Tp hybrid at different magnifications (I–III). **c** Confocal laser scanning microscopy (CLSM) images of the AgNC@ZnO Tp excited at 3.06 eV/405 nm; (I) showing a PL below/above 2.0 eV/615 nm, (II) bright-field image, and (III) the overlay of bright-field and PL intensity images. Reprinted with permission from ref. [82], Copyright 2021 The Royal Society of Chemistry. **d** Schematic illustration of the synthesis processes for (**a**) Au$_{25}$(SG)18@ZIF-8 (assembling of Au$_{25}$(SG)$_{18}$ into ZIF-8 matrix) and (**b**) Au$_{25}$(SG)$_{18}$/ZIF-8 (impregnating Au$_{25}$(SG)$_{18}$ on the outer surface of ZIF-8). **e** HAADF–STEM images of (I,II) 1000 Au$_{25}$(SG)$_{18}$@ZIF-8, and (I′,II′) 1000 Au$_{25}$(SG)18/ZIF-8. Reprinted with permission from ref. [83], Copyright 2017 WILEY.

interactions between the carboxyl groups of the AgNCs and Zn. The ZnO Tp (green luminescence) and AgNCs (red luminescence) show distinct luminescent characteristics under excitations at room temperature. Still, a noticeable change was observed in the formed hybrid AgNC@ZnO Tp (a faint yellow-orange luminescence) under the same excitation. Their luminescence properties vary with the excitation energy and temperature (Fig. 2c). A decrease in the luminescence of the nanohybrid was observed over time, limiting their applications. Luo et al[83]. synthesized stable Au$_{25}$(SG)$_{18}$–ZIF-8 nanohybrids (SG: glutathione) by intercalating the NCs into the ZIF-8 matrix (termed as Au$_{25}$(SG)$_{18}$@ZIF-8) or embellishing its outer surface (termed Au$_{25}$(SG)$_{18}$/ZIF-8) (Fig. 2d). In Au$_{25}$(SG)$_{18}$@ZIF-8 nanohybrids, Zn$^{2+}$ ions were coordinated with the carboxyl group of SG-ligand of Au NCs and the N atoms of 2-MeIM linkers, which resulted in the coordination-assisted self-assembly. But for the Au$_{25}$(SG)$_{18}$/ZIF-8 nanohybrids, NCs were integrated along the outer surface

of ZIF-8 by simple impregnation (Fig. 2e). Both the hybrids exhibit attractive photoluminescence properties and catalytic activities. Overall, in most of the cases, it was found that the surface properties of the resultant nanohybrids are primarily dependent on the types of ligands present on the NC's surface. Hence, tuneability in ligands structure might play a vital role in designing new nanohybrids.

### Class-II nanohybrids (hybridization via non-covalent interactions)

*Hybridization via hydrogen-bonding.* Hydrogen-bonding is a key reason for several self-assembly and hybrid particle formation, as it plays a commendable role in the thermodynamic stability of resultant particles. By suitably varying the reaction conditions, nanohybrids made of NCs can be created using H-bonds between the ligands on the NCs and the other nanoparticle's surface.

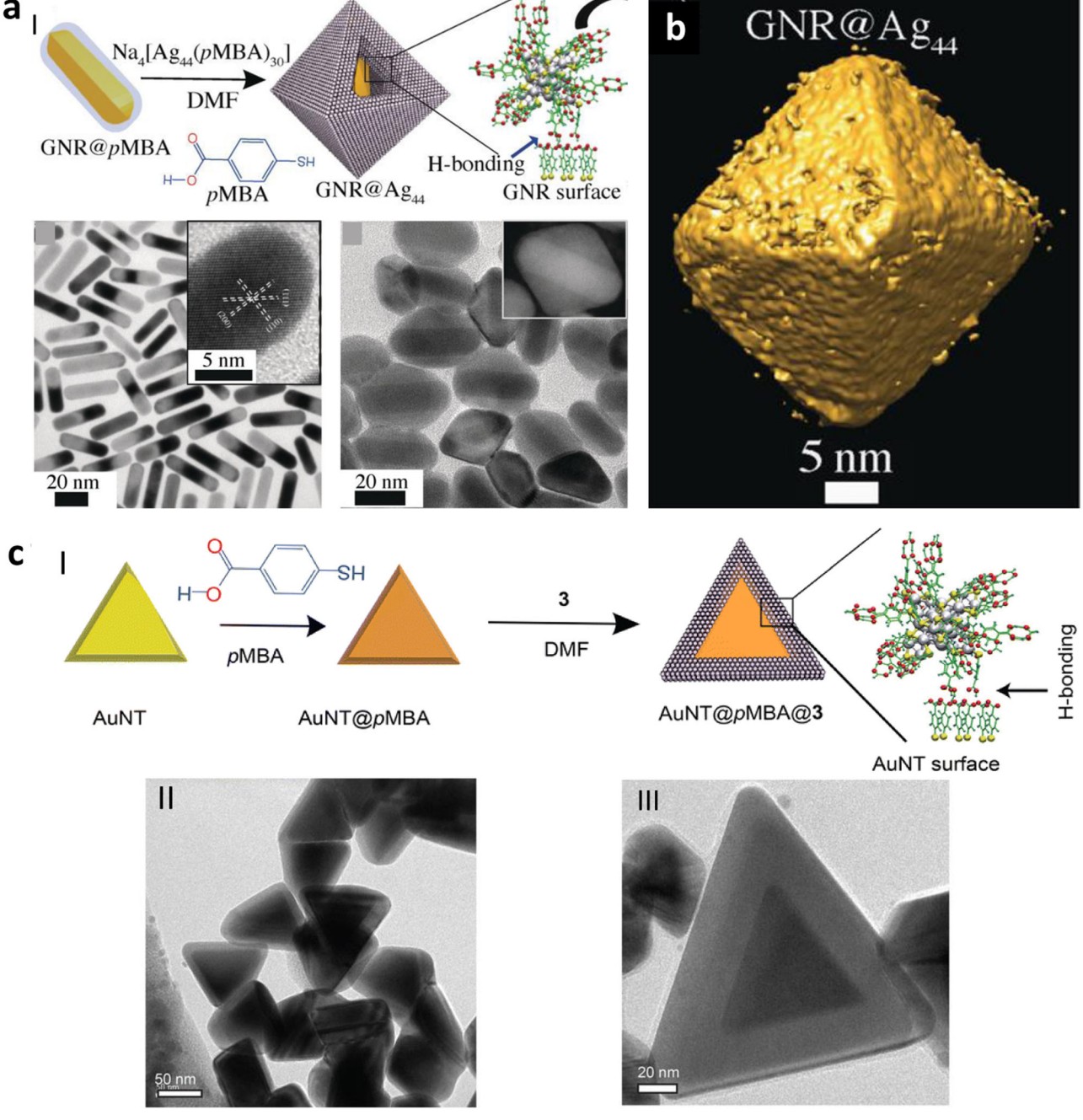

**Fig. 3 Nanohybrids based on hydrogen bonding interactions. a** (I) Representation of the self-assembly of Ag$_{44}$ (curved arrow directs to the ESI MS of the cluster) on the GNR@pMBA surface, (II) TEM image of GNR@pMBA (inset shows an HRTEM image of a single nanorod), (III) TEM image of GNR@Ag$_{44}$ (inset shows dark field STEM image of a composite particle). **b** TEM 3D reconstructed structure of GNR@Ag$_{44}$, showing octahedral geometry. Reprinted with permission from ref. [84], Copyright 2018 Wiley. **c** Interaction between AuNT@pMBA and Ag$_{44}$: (I) Schematic representation of the H-bond mediated assembly of Ag$_{44}$ on AuNT@pMBA template, (II) TEM image of AuNT@pMBA@Ag$_{44}$ after 24 h, (III) that of a single AuNT@pMBA@Ag$_{44}$ clearly showing core-shell composite structure. Reprinted with permission from ref. [85], Copyright 2023 Royal Society of Chemistry.

Pradeep and his co-workers[84] reported the synthesis of nanohybrids using atomically precise Ag$_{44}$(4-MBA)$_{30}$ NCs (MBA: mercapto benzoic acid) gold nanorods (GNRs). For this case, at first, the GNRs were surface modified with 4-MBA, where the thiol end of the MBA was attached to the gold of the GNR surface, and carboxyl groups were accessible. Next, Ag$_{44}$ (4-MBA)$_{30}$ was allowed to interact with the surface-modified GNRs via hydrogen-bonding between the two types of 4-MBA (one present on the surface of GNR and others are on the NC's surface). Standard features were observed in the absorption spectra with

the expected shift. Self-assembled Ag$_{44}$(4-MBA)$_{30}$ NCs were seen on the GNR surface in the corresponding TEM images (Fig. 3a). They have also reported a site-specific assembly of the NCs on the GNR surface. It was found that AgNCs predominantly prefer the (110) plane over (100) (anisotropic growth), which resulted in the nanohybrids of octahedral shape (Fig. 3b). In the presence of counter ions, it forms an isotropic FCC lattice held together by isotropic van der Waals interactions despite the disordered orientations of hydrogen bonds between connected NC units. These types of interaction provide extraordinary stability of the

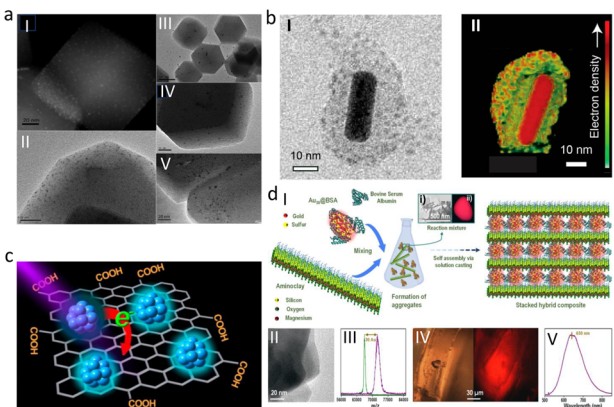

**Fig. 4 Nanohybrids based on electrostatic interactions. a** (I) HAADF-STEM image of the $Au_{12}Ag_{32}(SR)_{30}$@ZIF-8 composite; (II) TEM images of the $Au_{12}Ag_{32}(SR)_{30}$@ZIF-8 composite, (III) the $Au_{24}Ag_{46}(SR)_{32}$@ZIF-8 composite, (IV) the $Ag_{44}(SR)_{30}$@ZIF-8 composite, and (V) the $Ag_{12}Cu_{28}(SR)_{30}$@ZIF-8 composite. Reprinted with permission from ref. [86], copyright 2018 Royal Society of Chemistry. **b** (I) single AuNR@mSiO$_2$@Ag$_{29}$, and (II) a 3D reconstructed image of single AuNR@mSiO$_2$@Ag$_{29}$, where some of the atoms from the front part are omitted to give a clear core−shell view. Reprinted with permission from ref. [87], copyright 2022 American Chemical Society. **c** Illustration showing the luminescent Cu NCs-ImRGO nanocomposite. Reprinted with permission from ref. [88], copyright 2017 American Chemical Society. **d** (I) Schematic representation of the formation of the organo-inorganic CCH material. The inset shows the TEM image (i) and photograph (ii) of the intense red luminescent aggregates formed during the reaction of the clay and cluster. The schematic of the aminoclay sheet is presented for representation purposes only. (II) TEM image of the aminoclay sheet. (III) MALDI MS of the $Au_{30}$@BSA. (IV) Optical (left) and fluorescence (right) images of CCH. (V) Luminescence spectrum of CCH showed a maximum at 650 nm for 365 nm excitation. Reprinted with permission from Ref. [90], copyright 2021 American Chemical Society.

nanohybrids at room temperature. Since H-bonds are pH-dependent, the stability of nanohybrids decreased with the medium's pH increase.

In another work, Chakraborty et al.[85] addressed the role of ligands in determining the properties of resultant nanohybrids formed due to the interaction of AgNCs and gold nanotriangles (AuNTs). When AuNTs interacted with the $Ag_{25}(DMBT)_{18}$ NCs (DMBT:2,4-dimethylbenzenethiol), controlled etching was observed from the vertices and tips of AuNTs, whereas the formation of a dendritic nanoshell of Ag around the AuNTs was observed for $Ag_{25}H_{22}(DPPE)_8$ NCs (DPPE:1,2- bis(diphenylpho-sphino)ethane) and $Ag_{44}(pMBA)_{30}$ NC. But when the surface of the AuNT was modified with the 4-MBA ligands, similar to the report mentioned above, H-bonds driven self-assembly of Ag$_{44}$ NCs was seen on the AuNT surface, which resulted in a core-shell geometry where NTs were in the cores and NCs were assembled on the shell (Fig. 3c). Combination of the properties of plasmonic NPs and ultrasmall NCs, these nanohybrids could be effective surface-enhanced Raman active materials with ultrahigh sensitivities.

*Hybridization via electrostatic interactions.* Electrostatic self-assembly is an attempted method to produce well-blended nanohybrids[112]. Sun et al.[86] developed an 'electrostatic attraction strategy' to make encapsulated oil-soluble atomically precise NCs catalysts in a MOF (APNCs@MOF) via a "bottle around sheep" method. In this case, electrostatic interaction between $[Au_{12}Ag_{32}(SR)_{30}]^{4-}$ NCs [SR: $(SC_6H_3F_2)$] and cationic metal ions

of MOFs resulted in encapsulation of the NCs in the ZIF-8, ZIF-67, and manganese hexacyanoferrate hydrate (MHCF), respectively. Retention of structural integrity in the nanohybrids was confirmed from the UV-Vis spectra. The $Au_{12}Ag_{32}(SR)_{30}$@ZIF-8 has a uniform rhombic dodecahedral geometry, totally encapsulated inside the ZIF-8 matrix with excellent dispersity (Fig. 4a). When negatively charged NCs were replaced with positively charged NCs, almost no interaction was reported between the positively charged $Zn^{2+}$ of ZIF-8. The synergistic effect between ZIF and the NCs tremendously improves the nanohybrids' catalytic activity, combining both the components' advantages, i.e., the characteristic functional behavior of atomically precise NCs and the molecular sieving ability of the ZIF matrix. Pradeep and co-workers[87] reported the immobilization of negatively charged Ag$_{29}$ NCs on top of the silica-coated gold nanorods (positively charged), resulting a nanohybrids with plasmonic and fluorescent properties. AgNCs are anchored uniformly to each AuNR encapsulated. They can be characterized using tomographic reconstruction of the transmission electron micrographs (Fig. 4b). Maity et al.[88] synthesized the CuNC−ImRGO hybrid by electrostatic interaction of luminescent CuNCs ($Cu_7(L)_3$) with imidazole-functionalized partially reduced graphene oxide (ImRGO). In the $Cu_7L_3$ (L=cysteine) NCs, out of the three L-cysteine ligands, two prefer to remain as terminal ligands, while the third forms a $\mu_2$-bridging mode bond with the metal core. The cysteine-capped $Cu_7L_3$ system (Cys-CuNCs) is further stabilized by the primarily electrostatic, noncovalent interaction between copper and the oxygen center, which occurs over a minimal distance between them. Cys-CuNCs and ImRGO contain negatively charged carboxylate ions at the terminal of the CuNCs, and positively charged imidazole moieties present at the ImRGO layer respectively constructed the CuNC-ImRGO nanohybrids via electrostatic interaction. L-cysteine-capped CuNCs in an aqueous solution has a strong emission maximum at 489 nm and a quantum yield of 6.2%. Due to the presence of ImRGO, CuNCs' PL intensity was quenched by up to 87% as well as the PL lifetime was also decreased, which can be explained by the electron transfer process from CuNC to the lower energy levels of ImRGO (Fig. 4c). This electron transfer demonstrates charge separation in the excited state, which may pave the way for use in efficient light-harvesting systems. Yu et al.[89] constructed a sequence of hierarchical $Ni_xMg_{3-x}Al$-LDH/rGO ((LDH: layered double hydroxides, x = 1, 1.5 and 2, rGO: reduced graphene oxide) assisted atomically precise Au$_n$ NCs (n = 25, 38, and 127) catalysts $Au_n/Ni_xMg_{3-x}Al$-T (T = 270-300 °C). The nanohybrids were prepared to utilize atomically precise captopril-capped Au$_n$ NCs ($Au_{25}Capt_{18}$ NCs) as precursors using a double pH- controlled electrostatic adsorption strategy. In their report, the negatively charged $Au_{25}Capt_{18}$ NCs are adsorbed on the positively charged $Ni_{1.5}Mg_{1.5}Al$-LDH/rGO surface via the solid electrostatic attraction, producing the catalyst $Au_{25}Capt_{18}$/$Ni_{1.5}Mg_{1.5}Al$-LDH/rGO which was further subjected for proper calcination. In that catalyst, LDH nanosheets were grown vertically on both sides of rGO, and Au$_n$ NCs were positioned on edge and cross-connecting sites of the LDH/rGO hybrid, yielding an array-like nanosheet structure. All hierarchically constructed Au$_n$ NCs catalysts exhibited good catalytic activity for the base-free oxidation of benzyl alcohol. Catalytic activities displayed a rising trend with increasing Ni/Mg ratios and slowly decreasing with increasing the numbers of Au in Au$_n$ clusters. The catalysts have a high turn-over frequency (TOF) and exceptional reusability, making them useful in various catalytic applications.

In a recent report, Ghosh et al.[90] synthesized a highly luminescent organo-inorganic hybrid layered material with bovine serum albumin (BSA) protected $Au_{30}$ NCs and aminoclay sheets. Here, the coulombic interactions between the two

oppositely charged water-soluble $Au_{30}$@BSA NCs and the aminoclay sheets in solution triggered the formation of a water-insoluble hierarchical multilayered hybrid material (Fig. 4d). During the reaction, the clay-cluster hybrid (CCH) was first randomly oriented (Fig. 4d), and it then underwent self-assembly to create a multi-layered structure (Fig. 4d) where NCs were sandwiched between the two aminoclay layers. The CCH exhibits red luminescence under UV light, and its emission maxima near 650 nm (similar to $Au_{30}$ NCs), which indicates the NCs structure remains unchanged in the CCH (Fig. 4d). When the CCH was subjected to mechanical stresses like elongation, it underwent structural distortion that provided a new exciting optical feature (emission peak approximately 575 nm) because of the change in the Au-S bond length of $Au_{30}$ NC, which affected NCs' electronic states. This change in luminescence properties under some perturbations can be exploited in various mechanoresponses applications reported for other materials to generate pressure-based sensors[113,114].

**Class-III nanohybrids (hybridization via other interactions)**
*Hybridization via host-guest interactions.* Supramolecular assembly can modify the structure and shape of NCs and their characteristics which have been extensively studied for the past few years[14,115–118]. Recently, there has been a lot of interest in using macrocyclic hosts in host-guest assembly to create metal NC-based hybrid systems due to the high stability and interesting properties shown by these hybrid materials. Mathew et al.[91] introduced a convenient approach for making atomically precise hybrid NCs $Au_{25}SBB_{18} \cap CD_n$ (n = 1, 2, 3, and 4) using supramolecular host-guest interaction between β-cyclodextrin (CD) and 4-(t-butyl)benzyl (BBSH) mercaptan caged $Au_{25}$ NC. In this case, the cyclic oligosaccharide CD consists of seven α-D-glucopyranose units connected through α (1-4) glycosidic linkages that tend to bind selectively with particular hydrophobic guest species. BBSH ligands make host-guest solid interactions with the CD molecules, which increase the metallic core's stability by protecting the composite complex from various external ligands, metal ions, or other destabilizing agents. Matrix-assisted laser desorption ionization mass spectrometry (MALDI MS) analysis (Fig. 5a) of $Au_{25}SBB_{18} \cap CD_n$ with different SBB/CD ratios indicates successful complexation of β-CD on the surface of $Au_{25}$ NC as the shifting of peak maxima occurred towards the higher mass region with increasing CD concentration. Additionally, ~20 nm (Fig. 5b) in the absorption band was also observed in the UV-vis spectra. The $Au_{25}SBB_{18}$ NCs exhibit luminescence maxima at 1030 nm (NIR region) at room temperature, while the inclusion of β-CD enhanced the NIR luminescence intensity of the NCs (Fig. 5c) by reducing the non-radiative decays. This enhanced luminescence of the adduct remains intact even after 14 days, attributed to improved stability of the NCs due to the inclusion of CD. These surface-modified hybrid NCs are very effective for optoelectronic and sensor-based applications. Liu et al.[92] used atomically precise $Au_{133}(SR)_{52}$ NC (SR: SPh-p-Bu$^t$) as a guest species to form a nanohybrid with an isorecticular series of MOFs containing 3-D gradient porosity. The controlled synthesis of pores with different dimensions may be able to organize and transport molecular and nanoscale matter to particular locations in 3-D space. The NCs with the size of ~3 nm can be selectively incorporated into the periphery region of bMOF-106, not into the bMOF-102 due to the bigger size concerning the pore size (Fig. 5d) via host-guest interactions. Absorbance spectra of the MOF crystal after being treated with $Au_{133}(SR)_{52}$ [SR: 4-tert-butylbenzenethiol] reveal additional new characteristics of $Au_{133}(SR)_{52}$ (Fig. 5e) in the hybrids, and the UV-VIS spectral analysis of the fractured crystals exhibit that the

NCs located mainly in the peripheral area of the bMOF-102/106 (Fig. 5f). In another report Ganayee et al.[93] reported the enhancement of PL intensity of $Ag_{56}Se_{13}S_{15}$@$SBB_{28}$ NCs by forming the nanohybrids with aminoclay grafted with CD (AC-CD) via host-guest interactions. The host-guest complex was formed (Fig. 5g) between the $Ag_{56}Se_{13}S_{15}$@$SBB_{28}$ NC and β-cyclodextrin of AC-CD (aminoclay, grafted with CD). Here the functionalization of AC with monochlorotriazinyl (MCT) ligated β -CD yields AC-DC adduct, and its supramolecular assembly with the $Ag_{56}Se_{13}S_{15}$@$SBB_{28}$ NCs gives highly luminescent AC-CD∩$Ag_2$Se@SBB nanohybrid. The UV-vis spectra revealed a blue shift of absorption bands of AC-CD∩$Ag_2$Se@SBB compared to pristine metal nanocluster due to the supramolecular interaction between cluster and AC-CD. Although the nanohybrid's absorption feature is similar to the parent NCs, the photo-luminescence (PL) intensity was 1.4 times higher than the original NCs. In most cases, the enhanced Pl intensity is due to the surface rigidity, which minimizes the non-radiative decay.

Nag et al.[94] in 2018 synthesized the supramolecular adduct named as $[Ag_{29}(BDT)_{12} \cap (CD)_n]^{3-}$ (n = 1 – 6)[BDT = 1,3-benzene dithiol & CD = Cyclodextrins(α, β and γ)]. They demonstrated the phenomenon of isomerism in them. During adduct formation, various host-guest complexes were characterized in gas and solution phases using ion mobility mass spectrometry (IM MS). Detailed investigation on isomer identifications was also carried out using the IM MS, where it was found that $[Ag_{29}(BDT)_{12} \cap (CD)_n]^{3-}$ (n = 1-6)] exhibits isomers were only detected when n = 2, 3, and 4. Cis-trans isomers were found for n = 2,4, and fac-mer isomers were exhibited when n = 3. The distinct isomers are indicated by the drift time profile (Fig. 5h) of different supramolecular adducts. Here, hydrogen bonding, weak ionic Ag-O, van der Waals (vdWs) interactions, and C − H···π are the primary causes of adduct formation. It is observed that from α to γ CD, the Ag-O interaction becomes stronger. It can be seen in optimized structures that the trans isomers are less stable than cis isomers because the latter contains extra hydrogen bonds between the CDs. Density functional theory (DFT) and molecular docking and collision cross section (CCS) calculations were used for the structural understanding of these hybrids. Chakraborty et al.[95] presented the construction of supramolecular adducts between fullerenes and atomically precise $Ag_{29}(BDT)_{12}$ NCs, and the hybrid was termed as $[Ag_{29}(BDT)_{12}(C_{60})_n]^{3-}$ (n = 1 − 9). Fullerenes with high numbers of degenerate LUMOs, can receive multiple electrons, resulting in polyanionic species. $[Ag_{29}(BDT)_{12}(C_{60})_n]^{3-}$ (n = 1–4) were characterized entirely in the gaseous and solution phases. They were principally stabilized by van der Waals (VDW) forces and interactions between the fullerenes with the benzene rings present in the BDT ligands of the NCs. Three of the benzene rings of the BDT ligands are oriented to each of the cluster's tetrahedral vertex positions, improving the stabilization of a $C_{60}$ molecule through π − π interaction and C − H−π interactions at the ligand periphery, and the unpassivated Ag atom gets their stabilization by the $C_{60}$ through a $\eta^2$ interaction at 6-5 bond of the same. In collision-induced dissociation (CID) spectra (Fig. 5i, j) for $[Ag_{29}(BDT)_{12}(C_{60})_n]^{3-}$ (n = 2- 4) at collision energy (CE10), the sequential loss of neutral $C_{60}$ molecules from the cluster was observed with increase CE, providing the structural details of the hybrid. Such fullerene-cluster composites are fascinating materials for their properties (photoinduced charge transfer, electrical conductivity, etc.) and various applications in many fields like sensors, photovoltaics etc.

*Hybridization via hydrophobic interactions.* In the case of nano-hybrid materials made of hydrophilic/ hydrophobic NCs,

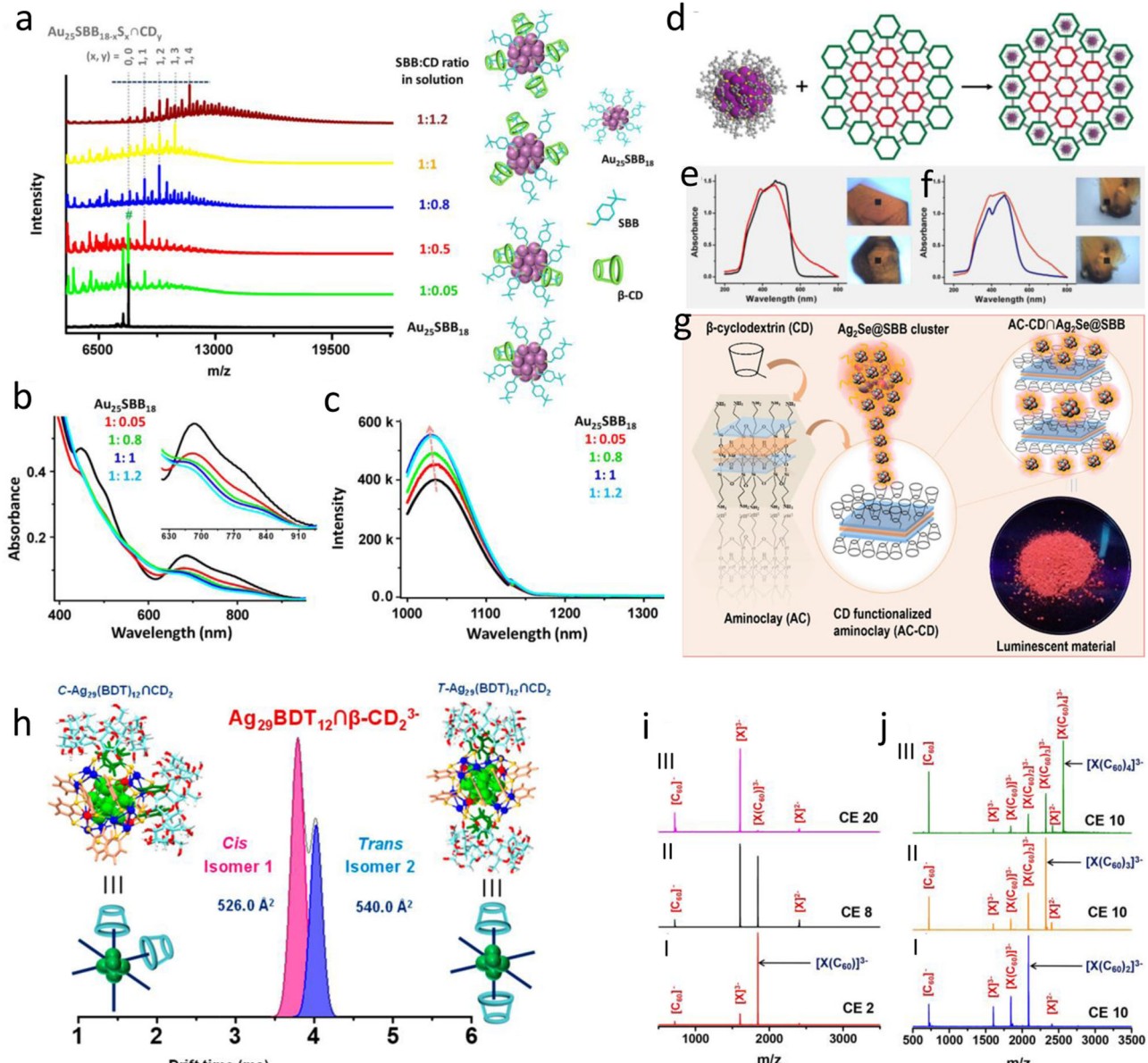

**Fig. 5 Nanohybrids based on host-guest interactions.** Effect of MALDI TOF TOF mass spectra of $Au_{25}SBB_{18}$ (black trace) with increasing SBB/CD ratio (green to brown trace) in solution. Schematic cluster representations with different amounts of CD inclusions are also shown. At 1:0.05, some parent $Au_{25}SBB_{18}$ is also seen, shown with #. UV-vis absorption spectra (**b**) and luminescence spectra ($\lambda_{ex} = 992$ nm) **c** of the $Au_{25}SBB_{18}$ cluster with increasing amounts of CD inclusion. Reprinted with permission from ref. [91], copyright 2013 American Chemical Society. **d** Schematic representation of "ideal" $Au_{133}(SR)_{52}$ organization in the periphery of gradient bMOF-102/106. **e** Absorbance spectra of bMOF-102/106 before (black) and after (red) encapsulation of $Au_{133}(SR)_{52}$ and corresponding optical images (before, top; after, bottom). **f** Absorbance spectra of different regions of a bMOF−102/106 crystal showing the presence of $Au_{133}(SR)_{52}$ in the periphery (orange) but not the core (blue) and corresponding optical images (periphery, top; core, bottom). Reprinted with permission from ref. [92], copyright 2016 American Chemical Society. **g** Schematic representation of entrapment of atomically precise clusters in cyclodextrin-functionalized aminoclay (AC-CD). Reprinted with permission from ref. [93], copyright 2020 American Chemical Society. **h** Drift time profile of ($n = 2$) complexes and its corresponding CCS value indicates the cis-trans isomers of $n = 2$. Reprinted with permission from ref. [94], copyright 2018 American Chemical Society. **i** CID study on $[X(C_{60})]^{3-}$ at increasing collision energies (CE in instrumental units) of (I), (II), and (III). **j** CID spectrum of $[X(C_{60})_n]^{3-}$ ($n = 2-4$) are shown in (I-III) respectively, at CE 10, X = $Ag_{29}(BDT)_{12}$. Reprinted with permission from ref. [95], copyright 2018 American Chemical Society.

hydrophobic interaction may provide additional potential force that might play a crucial role in determining their structures, stability, and properties. Hydrophobic interactions, also referred to as the hydrophobic effect, are a set of characteristics exhibited by nonpolar substances that lead to their self-assembly into anhydrous domains within polar solvents (aqueous solutions). The hydrophobic effect is primarily driven by the entropy factor caused by nonpolar solutes breaking hydrogen bonds between

water molecules[119,120]. Guan et al.[96] reported an effective exfoliation and functionalization method to synthesize hybrid Au/$MoS_2$ nanosheets using bovine serum albumin (BSA) protected $Au_{25}$ NCs and $MoS_2$ nanosheets (Fig. 6d). The nonpolar benzene rings of BSA have high binding energy with $MoS_2$ compared to the other polar groups of BSA, resulting in the adsorption of BSA molecules strongly on the surface of $MoS_2$ nanosheet through the hydrophobic interactions at optimum pH=4. This results in

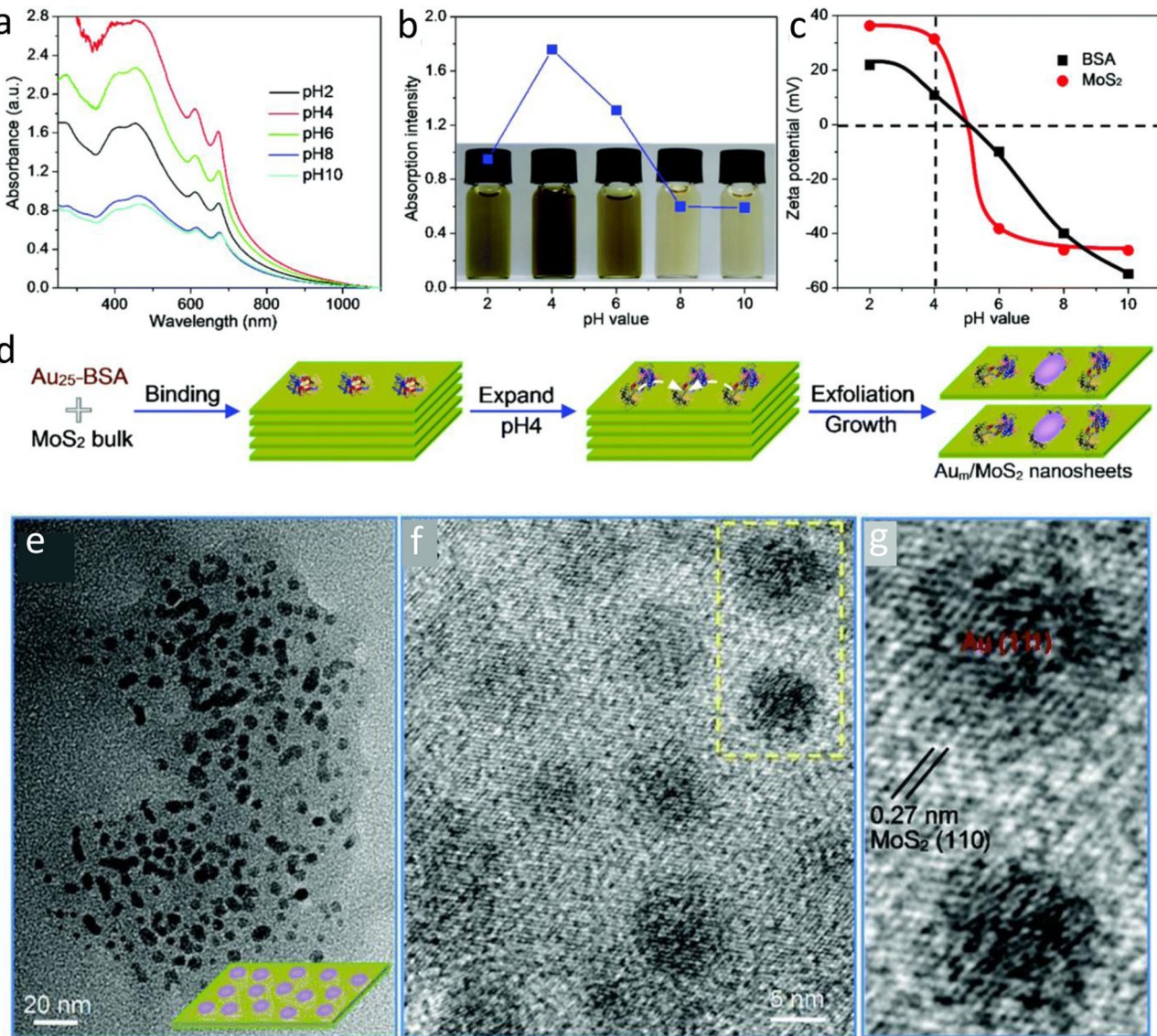

**Fig. 6 Nanohybrids based on hydrophobic interactions.** pH-Dependent exfoliation and hybridized functionalization of $MoS_2$ nanosheets with BSA-caged $Au_{25}$ clusters ($Au_{25}$-BSA). **a** UV-vis absorption spectra of $MoS_2$ nanosheets exfoliated at different pH values with BSA as an effective cleaving agent. **b** The evolution of absorption intensity of $MoS_2$ nanosheets as a function of pH together with the corresponding optical images. **c** Zeta potentials of $MoS_2$ and pure BSA in different pH environments. **d** Schematic exfoliation of $MoS_2$ nanosheets and subsequent surface growth of $Au_{25}$ clusters into $Au_m$ nanoparticles. **e** Low-resolution and **f** High-resolution TEM images of $Au_m/MoS_2$ nanosheets, and **g** enlarged TEM image of the highlighted area in **f**. Reprinted with permission from ref. [96], copyright 2018 Royal Society of Chemistry.

efficient exfoliation of the nanosheets along with the construction of 5 nm $Au_m$ NPs due to "epitaxial growth" of $Au_{25}$ cores. UV-vis absorption spectra and optical pictures beautifully explained the pH dependence of this exfoliation process (Fig. 6a-c) where the low-resolution TEM image provided the morphology of the (Fig. 6e) and the presence of 5 nm-sized $Au_m$ nanoparticles on the composite. In contrast, the

high-resolution TEM image (Fig. 6f, g) demonstrates the identical lattice spacing of 0.27 nm, belonging to the (111) plane of Au NPs and (110) plane of $MoS_2$ giving the evidence of the epitaxial growth of $Au_{25}$ nanoclusters. Besides, this hybridization significantly impacted the enhanced photocatalytic performance compared to those individual materials. Due to the smooth movement of the photogenerated holes and electrons across the $Au/MoS_2$ interface, the hybridized nanosheets degraded the methylene blue (MB) dye more quickly, and the degradation percentage increased significantly.

Applications of Nanohybrids: These nanohybrids exhibit synergistic/anti-synergistic properties and even some new properties depending on the type of interactions between the NCs and the other nanomaterials. This opens up the possibility to use such nanohybrids in a broad variety of novel applications starting from energy to bio-applications. This review will briefly summarize the potential uses of NCs-based nanohybrids in some emerging fields such as photovoltaics[121–124], photocatalysis[125–129], antimicrobials[130–135].

The interesting properties such as size dependent emission spectra, distinct absorption features, non-toxic behavior, scalable and ambient synthesis procedures of metal NCs make them a potentially useful material for photovoltaic applications, such as light-emitting diodes (LEDs). However, those NCs suffer from instability, low photoluminescence intensity, and a single distinguishable emission color, which also makes it challenging to use for such applications. However, nanohybrids consisting of

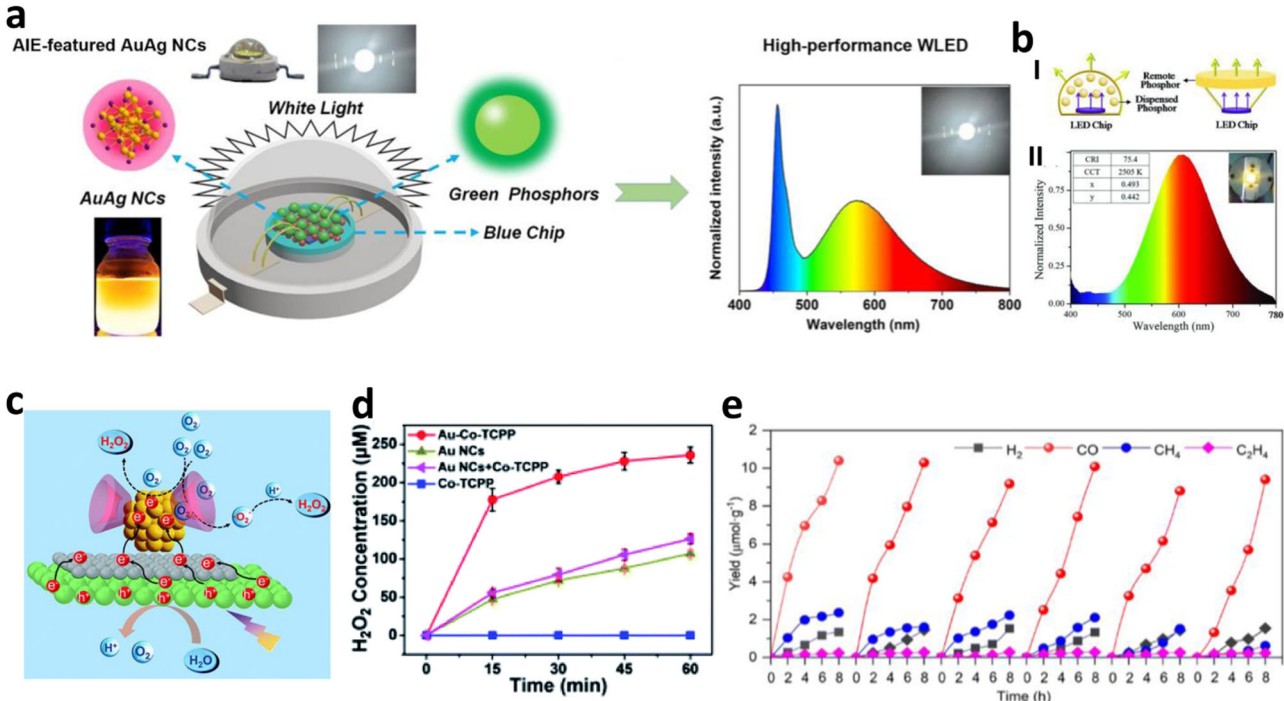

**Fig. 7 Nanohybrids for LEDs and photocatalysis applications. a** Schematic illustration of a representative WLED constructed by combining AIE-Featured orange-emitting AuAg NCs with green phosphors on a blue chip. Reprinted with permission from ref. [121], copyright 2020 American Chemical Society. **b** (I) Schematics of a conventional (left) and a remote type (right) down-conversion LED; (II) Emission spectrum of a remote LED employing the HGC/Cu NC film; inset on the left provides CRI, CCT, and CIE chromaticity coordinate of the device, and inset on the right shows a photograph of the working orange emitting device. Reprinted with permission from ref. [123], copyright 2018 WILEY-VCH. **c** proposed mechanism for photocatalytic $H_2O_2$ production over the Pt@b-CD/$C_3N_4$-M photocatalyst. Reprinted with permission from ref. [126], copyright 2021 Royal Society of Chemistry. **d** Time-course photocatalytic $H_2O_2$ production over Au-Co-TCPP, Au NCs + Co-TCPP, pristine Au NCs and Co-TCPP. Reprinted with permission from ref. [127], copyright 2022 Royal Society of Chemistry. **e** Light-driven catalytic durability over $Au_{25}$@Cu-BTC. Each cycle takes 8 h. Reprinted with permission from ref. [128], copyright 2023 Royal Society of Chemistry.

metal NCs and other functional nanomaterials can address those challenges and can be used for high-performance LED applications. For example, Yin et al.[121] reported a high-performance white light emitting diode (WLED) based on AuAg bimetallic NCs with a high CRI (80.6), high LE (91.9 lm/W), and nonchromaticity parameter drift. The bare AuAg bimetallic NCs have a bright orange emission (610 nm), large stokes shift, and a high PLQY (43% in solid-state), making them appropriate for high-end lighting applications. When orange emitting AuAg bimetallic NCs and green $Y_3(Al, Ga)_5O_{12}$:$Ce^{3+}$ phosphors (Ga-YAG phosphor) (emission maxima 530 nm) was combined with a molar ratio of 1:10 and integrated the mixture on a blue-emitting (emission maximum 455 nm) InGaN-based LED chip, the resulting hybrid material was able to function as a trichromatic white light-emitting diode (WLED) (Fig. 7a). Here the orange light emitted from the AuAg nanocrystals, green light from the Ga-YAG phosphor, and blue light from the InGaN chip were combined to produce white light. While, the correlated color temperature (CCT) of white light-emitting diodes (WLEDs) can be tailored to a desired range by adjusting the proportions of nanocrystals and Ga-YAG, producing warm light and more effective for human eyes and many other creatures. Bhandari et al.[122] also prepared bio-friendly luminescent white light emitting NC-based nanohybrid composed of red emitting AuNCs, green emitting $ZnQ_2$ complex (a complex of $Mn^{2+}$ doped ZnS Q dot and 8-hydroxyquinoline (HQ)) and blue emitting protein. Wang et al.[123] have successfully synthesized biocompatible HGC/CuNCs films with exceptional luminescence properties. The fabrication involved utilizing negatively charged CuNCs as a component of a positively charged biocompatible

graft copolymer, specifically hypromellose grafted chitosan (HGC). The HGC with a positive charge served a dual purpose in inducing the aggregation of CuNCs and serving as the matrix for developing films with exceptional luminescent properties (Fig. 7b). Compared to bare CuNCs, the resultant film exhibits an increase in PLQY from 0.5% to 42%, an improvement in PL average time from 6.1 μs to 25.9 μs, and a blue shift in the PL spectrum (649 nm to 600 nm). These films are used as color convertors to fabricate down conversion remote LED. Yang et al.[124] developed fluorescent GSH-CuNCs/Zn-HDS (hydroxy double salts) nanohybrid materials using surface CIEE (confinement induced enhanced emission) based on Zn-HDS as host material and GSH-CuNCs as guest molecules. The resulting hybrid exhibit high PLQY and longer fluorescence life time and higher stability than the bare CuNCs for LED applications. GSH-CuNCs/Zn-HDS powder was coated on the commercial UV LED chip provides orange emitting LED with luminance efficiency (1.5), color purity (85%), CCT (2461). The performance of this LED device against varying driving currents illustrates its potential for use in real-world application.

Atomically precise metal NCs with ultrasmall size emerge as efficient photocatalysts by virtue of their unique electronic and optical properties, high surface-to-volume ratio, and abundance of active sites[79,129]. Still, the photocatalytic performance has been affected by several limitations like light-driven aggregation of metal NCs, lack of active sites for catalysis, and quick recombination of the photogenerated electrons and holes[127,128]. Many studies have been done to improve photocatalytic efficiency by using NC-based hybrid materials. For example, Negishi et al.[125] demonstrated that hybridization of $BaLa_4Ti_4O_{15}$ with

tiny ($1.2 \pm 0.3$ nm) $Au_{25}(SG)_{18}$ NCs results in 2.6 times higher water splitting activity than hybridization of $BaLa_4Ti_4O_{15}$ with larger Au nanoparticles (10-30 nm),

which can be attributed to the distinct surface and electronic effects of atomically precise gold NCs. The gold NC acts as a co-catalyst and provides a more significant number of active sites for catalysis even with lower loading, resulting in increased catalytic activity. Zhu et al.[126] designed a hybrid photocatalyst- Pt@cyclodextrin NCs on $C_3N_4$/ MXene ($Ti_3C_2$) heterojunction (Pt@b-CD/$C_3N_4$-M) which delivers high $H_2O_2$ production of 147.1 μM $L^{-1}$ (~6 times greater $H_2O_2$ production with contrast to pristine $C_3N_4$). This excellent improvement in catalytic activity clearly reflects the complementary contribution of Pt cores where Pt acts as an active sites for catalysis and photogenerated e-acceptor, per-6-thio- β -cyclodextrin acts as hydrophobic delivery channel for $O_2$ transfer, $C_3N_4$ as photoinduced electron generator, Mxene as improved visible light absorber (Fig. 7c). Similarly, Xue et al.[127] reported a covalently hybridized Au-Co-TCPP hybrid photocatalyst which provides 2.2 times higher $H_2O_2$ production capability than the pure AuNCs by leveraging the synergistic contribution of the hybrid structure (Fig. 7d). Au-Co-TCPP exhibits a $H_2O_2$ production of 235.93 mM in 60 min, whereas the pristine AuNCs produce only 107.11 mM of $H_2O_2$. Here, in addition to improving visible light absorption of AuNCs, the grafted Co-TCPP unit may operate as electron acceptors, thereby effectively enhancing the charge separation of AuNCs while also offering an abundant supply of active sites for photocatalysis. As the recyclability or reusability of a catalyst is directly connected to the stability of the catalyst, encapsulation of metal NC or hybridization with a supporting framework is a very effective way to enhance the stability of the photocatalyst as well as addressing the issue of recyclability. So, Zhang et al.[128] designed a photocatalyst by encapsulating $Au_{25}$(p-MBA)$_{18}$ (p-MBA = 4-mercaptobenzoic acid) in a $Cu_3(BTC)_2$ (BTC = benzene-1,3,5-tricarboxylate) metal–organic framework ($Au_{25}$@Cu-BTC) which prevents the aggregation of AuNCs. The catalyst exhibits steady performance for successive 6 cycles (48 h), representing its outstanding stability (Fig. 7e). Besides, "covalence bridge' between the two components of the hybrid NC is an effective strategy to boost both the activity and durability[129].

Recently, such ultrasmall metal NCs with core size of ≤3 nm have been demonstrated to be a new type of effective antibacterial agent[133,135]. Excellent photodynamic antibacterial efficacy of metal NC-based photocatalysts demands good photostability, outstanding harvesting ability for visible light, and superior productivity/separation of sufficient charge carriers. Interestingly, several types of AgNCs and AuNCs have been successfully designed as wide-spectrum antibacterial agents with improved bacterial killing effect. Such as, Wang et al.[130] developed a AgNCs@CH-MF hybrid hydrogel (MF: mangiferin, CH – Chitosan) based on AgNCs and CH-MF, which shows significant synergistic and multiple impacts on bactericidal performance. Besides interfacial design of the hydrogel, the AgNCs@CH-MF hydrogel's strong antibacterial efficacy, was further improved due to the self-degradation into microparticles. AgNCs@CH-MF hybrid hydrogel possessed excellent biocompatibility by enhancing cell proliferation and prompting tissue regeneration. The AgNCs@CH-MF hydrogel has many advantages such as injectability, adequate swelling, good degradability, and good controlled release properties and shows antibacterial activity against both gram-positive and gram-negative bacteria by killing the adhered bacteria on the hydrogel surface with a high local concentration of Ag species. Similarly, Zheng et al.[131] synthesized the MXene-AuNCs conjugated system which achieves excellent synergistic antimicrobial activity as MXenes pierce the bacterial membrane and conjugated AuNCs would be better internalized

inside bacteria to generate reactive oxygen species (ROS) for disrupting bacterial normal metabolism. Furthermore, the MXene nanosheets can also induce oxidative stress in bacteria, the resulting ROS are concentrated locally and function as a ROS reservoir, allowing for continual oxidation of bacterial membrane lipid for accelerated membrane breakdown and bacterial DNA for violent fragmentation, which ultimately results in the death of bacteria. In another work, Wang et al.[132]. have developed AgNCs@ELB (ELB: extract of Luria-Bertani (LB) medium) nanohybrid with antibacterial characteristics through the encapsulation of AgNCs within the sacrificial ELB species using a facile light irradiation process. The AgNCs@ELB, upon being swallowed by the bacteria, could be effectively triggered once its ELB shell was digested by the bacteria. The as-designed AgNCs@ELB is highly efficient and biocompatible because of the synergistic effects of sacrificial ELB vehicle together with AgNCs[132]. Liu et al.[133] demonstrated CDs@AuAg NCs (CD: carbon dot), exhibited enhanced photodynamic antibacterial performance (Fig. 8a), as conjugation of CD with AuAg NCs could enhance the harvest of visible light also ROS production and because of the small size and good water-solubility, CDs@AuAg NCs could promote their interaction with bacteria. The probable mechanism suggested by the authors is that upon visible-light illumination, the CDs' electrons might be excited, and some of them might then move from the conduction band (CB)level of the CDs to the LUMO level of the AuAg NCs through a charge transfer route, facilitating their surface photocatalytic interaction with dissolved oxygen ($O_2$) to produce ROS. Zhu et al.[134] prepared $TiO_2$-$NH_2$@AuNC (Fig. 8b) nanohybrid in which the chemically grafted AuNCs on the $TiO_2$-$NH_2$ surface can serve as a high-efficiency photosensitizer to harvest visible light to promote the charge carriers ($e^-$/$h^+$ pairs) generation. After that, the photoexcited $e^-$ would transfer from the excited AuNCs to the CB of $TiO_2$ via amide bonds and finally migrate to the surface of $TiO_2$, reacting with the dissolved oxygen ($O_2$) for the production of ROS. In addition, the electron-rich amino groups on the $TiO_2$-$NH_2$ surface can further enhance their surface charge density via charge transfer, eventually promoting the production of ROS, resulting in an excellent photodynamic antibacterial agent. The antibacterial activity can be explained by the highly luminescent AuNCs acting as a visible-light sensitizer for the photodynamic killing of bacteria, and the electrostatic interaction between $TiO_2$ and the AuNCs may make some of them as photocatalytic entities and facilitate the generation of ROS for killing the bacteria (Fig. 8c). Zheng et al.[135] reported a unique magnetically oriented Ho-GO-Au nanohybrid system which utilizes both physical (via oriented GO) and chemical (via GO and AuNCs) mechanisms to perform as a bacterial killing agent. Under weak magnetic fields, the Ho-GO-Au nanosheets were able to be vertically orientated, providing high-density sharp edges with favored orientation to successfully damage bacterial membranes. The conjugated AuNCs were efficiently delivered into GO-cut bacteria and induced high oxidative stress, which strongly disturbed bacterial metabolism, leading to the death of the bacteria.

**Summary and outlook**. In summary, the recent progress on nanohybrids made of atomically precise NCs was categorized based on the type of interactions between the NCs and other nanomaterials. Though atomically precise metal NCs have exciting physical/chemical properties, including PL, and catalysis, their long-term instability under ambient atmosphere creates a significant issue in application fields. Hybridization with other nanomaterials can significantly improve their stabilities and make them useful in diverse applications. The hybrid materials were categorized into classes based on the types of interactions (such as

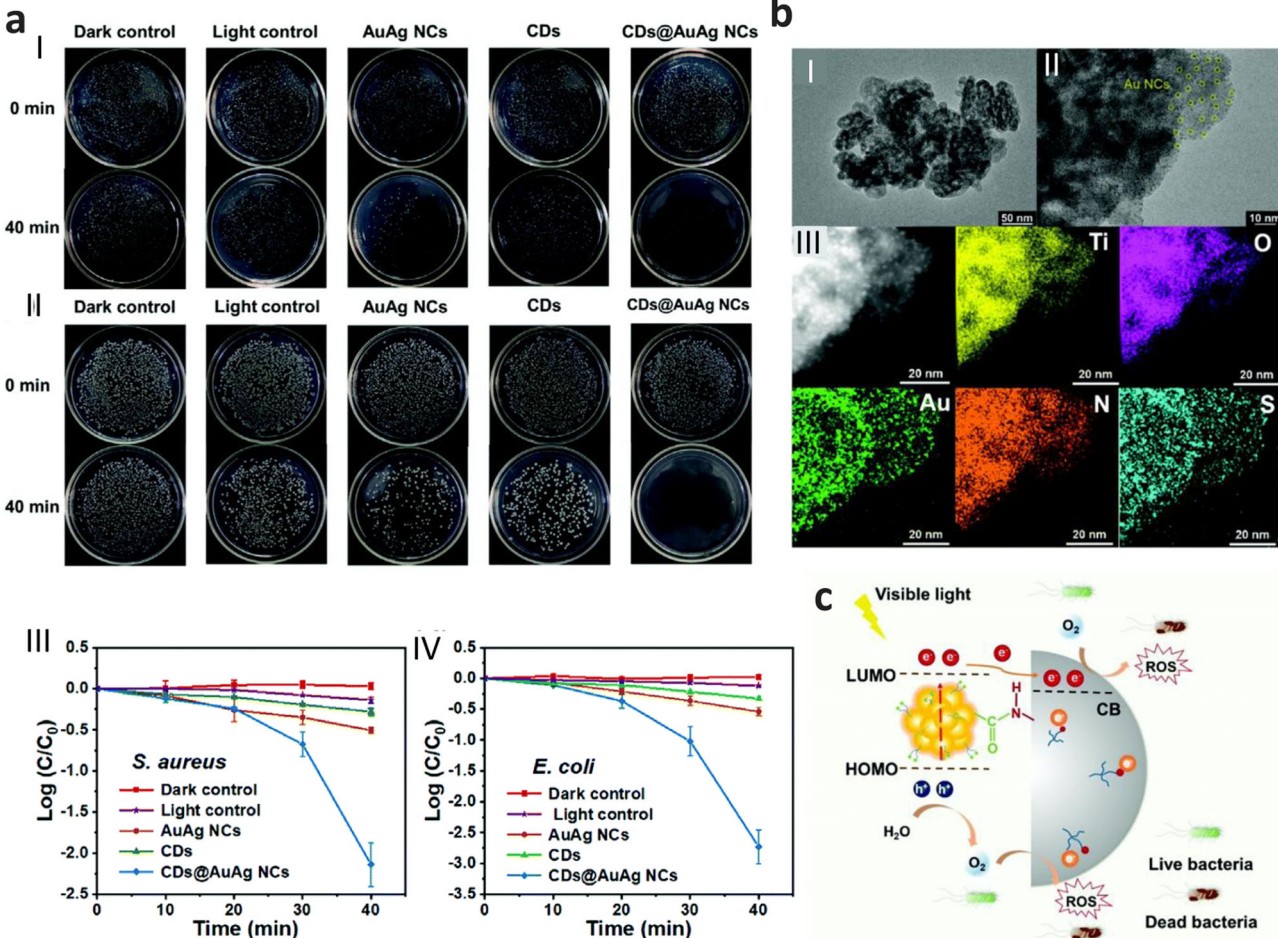

**Fig. 8 Nanohybrids for antibacterial applications. a** Bacterial colony growth of (I) S. aureus and (II) E. coli in the presence of AuAg NCs, CDs, CDs@AuAg NCs, and water (as control) under visible-light illumination or dark conditions for 40 mins bactericidal efficiencies of (III) Gram-positive S. aureus and (IV) Gram negative E. coli treated with AuAg NCs, CDs, CDs@AuAg NCs, and water (as control) under visible-light illumination or dark conditions. Reprinted with permission from ref. [133], copyright 2022 Royal Society of Chemistry. **b** (I and II) TEM images of the $TiO_2$-$NH_2$@AuNC antibacterial agent with different magnifications. (III) EDS mapping image of the $TiO_2$- $NH_2$@Au NCs and the corresponding elemental mappings of Ti, O, Au, N, and S elements. **c** The proposed mechanism of the visible-light-driven antibacterial process of $TiO_2$-$NH_2$@AuNCs. Reprinted with permission from ref. [134], copyright 2021 Royal Society of Chemistry.

covalent (I), non-covalent (II), and other interactions (III), respectively). The nanohybrids exhibited either a synergistic effect, an anti-synergistic effect, or new properties from their components. Exciting properties such as PL enhancement, enhanced photocatalytic activities, enhanced stabilities, and high catalytic activities were prominent in the nanohybrids. Only a handful of examples were highlighted in this perspective, but the field has just started evolving, and there are many opportunities to explore. To expand the NC-based nanohybrid canvas to other materials, it is important to employ new synthetic strategies that could fabricate various nanohybrids. Especially solid-state based reactions, high-pressure reactions, and organic coupling reactions (apart from EDC) could be used to form new nanohybrids with metal NCs. By controlling the external kinetic parameters, it is also necessary to precisely control the number of NCs involved in such nanohybrids. So far, most nanohybrids have been reported to have single atomically precise NCs. However, adding one or two different NCs (heterogeneity in atomicity, types of metals, or ligands) can significantly improve nanohybrid properties and make them more multifunctional. Most often, precise character-izations were not looked at in detail, such as the number of NCs or the number of other nanomaterials in each of the nanohybrids,

which play a significant role in determining their properties. For example, the number of NC present on the nanohybrids may affect the catalytic properties of the nanohybrids and the resultant turnover number calculations. So, it is crucial to characterize such nanohybrids in a precise way. High-resolution mass spectrometry and many advanced mass spectrometry methods can be imple-mented for their precise characterizations. To understand the surface topology of the nanohybrids, tomographic reconstruction of the transmission electron micrographs can be used together with elemental mapping. The nanohybrids' reports mainly focused on synthesizing or investigating their properties. Detail understanding of their formation (kinetics and dynamics) should have been explored in each case. The dynamics and kinetics can help us understand their formation mechanism, enabling us to design new nanohybrids. So far, these nanohybrids are used in LEDs, photocatalysis, and bio-applications such as antibacterial but recent understanding about these materials, especially their improved PL properties, can enable them to be used in potential sensor devices for health, environmental and sustainable solu-tions of life. Some of their potential applicability would be water purification, hydrogen generation, and medical diagnosis of rare diseases.

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

## Acknowledgements

The authors acknowledge the financial support for this work offered by Science & Engineering Research Board (SERB), project id: SRG/2022/000135, and SRIC, IIT Kharagpur (FSRG grant, project code DLR). K.S. thanks IIT Kharagpur for his research fellowship. T.R.G. thanks PMRF for his research fellowship. S.R. thanks UGC for his research fellowship as JRF.

## Author contributions

I.C. initiated the project. K.S., T.R.G., and S.R. performed an extensive literature study under the supervision of I.C. All authors co-wrote the manuscript.

## Competing interests

The authors declare no competing interests. Indranath Chakraborty is an Editorial Board Member for *Communications Chemistry*, but was not involved in the editorial review of, or the decision to publish this article.
