## [Peer Review File · Communications Chemistry]

Reviewers' comments:

Reviewer #1 (Remarks to the Author):

Metal nanoclusters-based hybrids are attractive due to their synergistic properties, and thus commonly constructed to address the challenging issues of various fields. In this perspective, the authors summarized the interaction types between the metal nanoclusters and substrates, and highlighted some representative metal nanoclusters-based hybrids with interesting properties. Generally, the perspective is timely, and of broad interest. Therefore, I strongly support its publication after a minor revision. Please see my detailed comments below.

- 1) While the authors stated “So far, there are no reviews or perspectives which highlight these nanohybrids, and this is probably high time to summarize the current understanding of this topic.”, I remember that Prof. Li Shang has published a review article with the topic of metal nanoclusters-based composites (<https://doi.org/10.1016/j.nantod.2019.100767>), and thus suggest the authors to revise their expression properly.
- 2) As mentioned by the authors, hybridization between metal nanoclusters and other nanomaterials often generate some new properties or improved application performance, which is the advantage of such hybrids. For example, metal nanoclusters-based hybrids have been demonstrated to be excellent in photocatalysis (<http://dx.doi.org/10.1039/D0TA10742E>; <http://dx.doi.org/10.1039/D2TA00720G>; <http://dx.doi.org/10.1039/C3NR01888A>), bacterial killing (<https://doi.org/10.1002/adhm.202001007>; <https://onlinelibrary.wiley.com/doi/abs/10.1002/adfm.201904603>; <https://doi.org/10.1007/s12274-019-2598-y>; <https://doi.org/10.1016/j.cej.2020.127589>; <http://dx.doi.org/10.1039/D1NR05503H>; <http://dx.doi.org/10.1039/D2NR01550A>), LED (<https://doi.org/10.1021/acssuschemeng.0c05722>; <https://onlinelibrary.wiley.com/doi/abs/10.1002/adfm.201802848>) and other applications (<https://doi.org/10.1021/jacs.0c00378>; <https://doi.org/10.1021/jacs.9b11017>; <https://doi.org/10.1021/acs.est.0c00427>) with intriguing performance. Therefore, in addition to discussing the interaction types, the interesting applications of metal nanoclusters-based hybrids should be briefly discussed to further enhance the broad interest to audience.
- 3) The authors stated that “but very few have been characterized by single x-ray crystallography, such as ...”, but now many hydrophobic thiolated ligands-protected Au nanoclusters have been characterized by this technique. Therefore, a suitable revision in the expression is expected to avoid possible confusion or misunderstanding.
- 4) Several typos, and format issues are observed in the context and Reference section.

Reviewer #2 (Remarks to the Author):

Review on Manuscript COMMSCHEM-23-0119

The Communication is a very well edited summary article. It summarises the background to the topic well.

My small comment is that some of the figures are quite small, the captions are not visible, which makes it difficult for the reader to understand the data. please enlarge the size of the very small letters.

One other small comment: a short thought on the quantum efficiency and lifetime data of fluorescent noble metal nanoclusters at the beginning of the article would be nice.

After these minor corrections I recommend the publication of this nice work.

Reviewer #3 (Remarks to the Author):

This work summarized the synthesis of nanohybrid based on the interaction between NCs and supports, but not mentioned the properties and application. The potential application of nanohybrid is more important to guide the oriented synthesis.

1. The proportion of forward looking and/or speculative is lesser than review in the whole text.

2. The category of nanohybrid is kind of unreason. It can be classified by the type of interaction or the type of support et al. The interaction could be covalence bond and non-covalence bond (electrostatic interaction, pi-pi interaction, hydrogen bond, and other Van der Waals intermolecular force et al.)

In the text, the host-guest interaction is a big scope including both the covalence and non-covalence interaction, while the hydrophobicity is more a property but not an interaction force.

3. The author name of ref. 88 in the text is inconsistent with the reference.

Therefore, some revision is needed and it is not suitable for publication as it is.

Response Letter

We appreciate all the reviewers for their precious time reviewing our paper and providing valuable comments. Your valuable and insightful comments led to possible improvements in the current version. We have carefully considered the comments and tried our best to address every one of them. Below we provide the point-by-point responses. The comments, corresponding responses, and the new text in the revised manuscripts are marked as black, blue, and green, respectively.

Reviewer #1

Comment 1.

While the authors stated “So far, there are no reviews or perspectives which highlight these nanohybrids, and this is probably high time to summarize the current understanding of this topic.”, I remember that Prof. Li Shang has published a review article with the topic of metal nanoclusters-based composites (<https://doi.org/10.1016/j.nantod.2019.100767>), and thus suggest the authors to revise their expression properly.

Response: We thank the reviewer for his/her generous suggestion. Based on that, we have modified the text in the revised manuscript. All the suggested references are added in the revised manuscript.

“So far, only a few reviews exist on nanocluster-based materials (especially composites), mostly on the synthesis and application aspects^{76, 77, 78, 79}.”

76. Yang J, *et al.* Metal nanocluster-based hybrid nanomaterials: Fabrication and application. *Coord. Chem. Rev.* **456**, 214391 (2022).
77. Bain D, Maity S, Patra A. Opportunities and challenges in energy and electron transfer of nanocluster based hybrid materials and their sensing applications. *Phys. Chem. Chem. Phys.* **21**, 5863-5881 (2019).
78. Lopes RCFG, Rocha BGM, Maçôas EMS, Marques EF, Martinho JMG. Combining metal nanoclusters and carbon nanomaterials: Opportunities and challenges in advanced nanohybrids. *Adv. Colloid Interface Sci.* **304**, 102667 (2022).
79. Shang L, Xu J, Nienhaus GU. Recent advances in synthesizing metal nanocluster-based nanocomposites for application in sensing, imaging and catalysis. *Nano Today* **28**, 100767 (2019).

Comment 2.

As mentioned by the authors, hybridization between metal nanoclusters and other nanomaterials often generate some new properties or improved application performance, which is the advantage of such hybrids. For example, metal nanoclusters-based hybrids have been demonstrated to be excellent in photocatalysis

(<http://dx.doi.org/10.1039/D0TA10742E>; <http://dx.doi.org/10.1039/D2TA00720G>; <http://dx.doi.org/10.1039/C3NR01888A>), bacterial killing (<https://doi.org/10.1002/adhm.202001007>; <https://onlinelibrary.wiley.com/doi/abs/10.1002/adfm.201904603>; <https://doi.org/10.1007/s12274-019-2598-y>; <https://doi.org/10.1016/j.cej.2020.127589>; <http://dx.doi.org/10.1039/D1NR05503H>; <http://dx.doi.org/10.1039/D2NR01550A>), LED (<https://doi.org/10.1021/acssuschemeng.0c05722>; <https://onlinelibrary.wiley.com/doi/abs/10.1002/adfm.201802848>) and other applications (<https://doi.org/10.1021/jacs.0c00378>; <https://doi.org/10.1021/jacs.9b11017>; <https://doi.org/10.1021/acsc.est.0c00427>) with intriguing performance. Therefore, in addition to discussing the interaction types, the interesting applications of metal nanoclusters-based hybrids should be briefly discussed to further enhance the broad interest to audience.

Response: Thank you very much for this suggestion. We have added a separate paragraph on the application aspect of the hybrid materials in our revised manuscript, and all the suggested references are added.

Applications of Nanohybrids:

These nanohybrids exhibit synergistic/anti-synergistic properties and even some new properties depending on the type of interactions between the NCs and the other nanomaterials. This opens up the possibility to use such nanohybrids in a broad variety of novel applications starting from energy to bio-applications. This review will briefly summarize the potential uses of NCs-based nanohybrids in some emerging fields such as photovoltaics^{121, 122, 123, 124}, photocatalysis^{125, 126, 127, 128, 129}, antimicrobials^{130, 131, 132, 133, 134, 135}.

The interesting properties such as size dependent emission spectra, distinct absorption features, non-toxic behaviour, scalable and ambient synthesis procedures of metal NCs makes them potentially material for photovoltaic applications, such as light-emitting diodes (LEDs). However, those NCs suffer from instability, low photoluminescence intensity, and a single distinguishable emission colour, which also makes it challenging to use for such applications. However, nanohybrids consisting of metal NCs and other functional nanomaterials can address those challenges and can be used for high-performance LED applications. For example, Yin et al. reported a high-performance white light emitting diode (WLED) based on AuAg bimetallic NCs with a high CRI (80.6), high LE (91.9 lm/W), and nonchromaticity parameter drift¹²¹. The bare AuAg bimetallic NCs have a bright orange emission (610nm), large Stokes shift, and a high PLQY (43% in solid-state), making them appropriate for high-end lighting applications. When orange emitting AuAg bimetallic NCs and green Y₃(Al, Ga)₅O₁₂:Ce³⁺ phosphors (Ga-YAG phosphor) (emission maxima 530nm) was combined with a molar ratio of 1:10 and integrated the mixture on a blue-emitting (emission maximum 455 nm) InGaN-based LED chip, the resulting hybrid material was able to function as a trichromatic white light-emitting diode (WLED) (Fig. 7a). Here the orange light emitted from the AuAg nanocrystals, green light from the Ga-YAG phosphor, and blue light from the InGaN chip were combined to produce white light. While, the correlated colour temperature (CCT) of white light-emitting diodes (WLEDs) can be tailored to a desired range by adjusting the proportions of nanocrystals and Ga-YAG, producing warm light and more effective for human eyes and many other creatures. Bhandari et al. also prepared bio-friendly luminescent white light emitting NC-based nanohybrid composed of red emitting Au NCs, green

emitting ZnQ₂ complex (a complex of Mn²⁺ doped ZnS Qdot and 8- hydroxyquinoline (HQ)) and blue emitting protein¹²². Wang et al. have successfully synthesized biocompatible HGC/CuNCs films with exceptional luminescence properties¹²³. The fabrication involved utilizing negatively charged CuNCs as a component of a positively charged biocompatible graft copolymer, specifically hypromellose grafted chitosan (HGC) The HGC with a positive charge served a dual purpose in inducing the aggregation of CuNCs and serving as the matrix for developing films with exceptional luminescent properties (Fig. 7b). Compared to bare CuNCs, the resultant film exhibits an increase in PLQY from 0.5% to 42%, an improvement in PL average time from 6.1 μs to 25.9 μs, and a blue shift in the PL spectrum (649nm to 600nm). These films are used as color converters to fabricate down conversion remote LED. Yang et al. developed fluorescent GSH-CuNCs/Zn-HDS (Hydroxy double salts) nanohybrid materials using surface CIEE (Confinement induced enhanced emission) based on Zn-HDS as host material and GSH-CuNCs as guest molecules¹²⁴. The resulting hybrid exhibit high PLQY and longer fluorescence life time and higher stability than the bare CuNCs for LED applications. GSH-CuNCs/Zn-HDS powder was coated on the commercial UV LED chip provides orange emitting LED with luminance efficiency (1.5), color purity (85%), CCT (2461). The performance of this LED device against varying driving currents illustrates its potential for use in real-world application.

Atomically precise metal NCs with ultrasmall size emerge as efficient photocatalysts by virtue of their unique electronic and optical properties, high surface-to-volume ratio, and abundance of active sites^{79, 129}. Still, the photocatalytic performance has been affected by several limitations like light-driven aggregation of metal NCs, Lack of active sites for catalysis, and quick recombination of the photogenerated electrons and holes^{127, 128}. Many studies have been done to improve photocatalytic efficiency by using NC-based hybrid materials. For example, Negishi et al. demonstrated that hybridization of BaLa₄Ti₄O₁₅ with tiny (1.2±0.3 nm) Au₂₅(SG)₁₈ NCs results in 2.6 times higher water splitting activity than hybridization of BaLa₄Ti₄O₁₅ with larger Au nanoparticles (10-30 nm), which can be attributed to the distinct surface and electronic effects of atomically precise gold NCs. The gold NC acts as a co-catalyst and provides a more significant number of active sites for catalysis even with lower loading, resulting in increased catalytic activity. Zhu et al. designed a hybrid photocatalyst-Pt@cyclodextrin NCs on C₃N₄/MXene (Ti₃C₂) heterojunction (Pt@b-CD/C₃N₄-M) which delivers high H₂O₂ production of 147.1 μM L⁻¹ (~6 times greater H₂O₂ production with contrast to pristine C₃N₄)¹²⁶. This excellent improvement in catalytic activity clearly reflects the complementary contribution of Pt cores where Pt acts as an active sites for catalysis and photogenerated e⁻ acceptor, per-6-thio- β -cyclodextrin acts as hydrophobic delivery channel for O₂ transfer, C₃N₄ as photoinduced electron generator, Mxene as improved visible light absorber (Fig. 7c). Similarly, Xue et al. reported a covalently hybridized Au-Co-TCPP hybrid photocatalyst which provides 2.2 times higher H₂O₂ production capability than the pure Au NCs by leveraging the synergistic contribution of the hybrid structure (Fig. 7d). Au-Co-TCPP exhibits a H₂O₂ production of 235.93 mM in 60 min, whereas the pristine Au NCs produce only 107.11 mM of H₂O₂¹²⁷. Here, in addition to improving visible light absorption of AuNCs, the grafted Co-TCPP unit may operate as electron acceptors, thereby effectively enhancing the charge separation of Au NCs while also offering an abundant supply of active sites for photocatalysis. As the recyclability or reusability of a catalyst is directly connected to the stability of the catalyst, encapsulation of metal NC or hybridization with a supporting framework is a very effective way to enhance the stability of the photocatalyst as well as addressing the issue of recyclability. So, Zhang et al. designed a photocatalyst by encapsulating Au₂₅(p-MBA)₁₈ (p-MBA = 4-mercaptobenzoic acid) in a Cu₃(BTC)₂ (BTC = benzene-1,3,5-tricarboxylate) metal-organic framework (Au₂₅@Cu-BTC) which prevents the aggregation of AuNCs¹²⁸. The catalyst exhibits steady performance for successive 6 cycles (48 h), representing its outstanding stability (Fig. 7e). Besides, “covalence bridge” between the two components of the hybrid NC is an effective strategy to boost both the activity and durability (Fig. 7f)¹²⁹.

Fig. 7 **a** Schematic illustration of a representative WLED constructed by combining AIE-Featured orange-emitting AuAg NCs with green phosphors on a blue chip. Reprinted with permission from ref.¹²¹. copyright 2020 American Chemical Society. **b** (I) Schematics of a conventional (left) and a remote type (right) down-conversion LED; (II) Emission spectrum of a remote LED employing the HGC/Cu NC film; inset on the left provides CRI, CCT, and CIE chromaticity coordinate of the device, and inset on the right shows a photograph of the working orange emitting device. Reprinted with permission from ref.¹²³. copyright 2018 WILEY-VCH. **c** proposed mechanism for photocatalytic H₂O₂ production over the Pt@b-CD/C₃N₄-M photocatalyst. Reprinted with permission from ref.¹²⁶. copyright 2021 Royal Society of Chemistry. **d** Time-course photocatalytic H₂O₂ production over Au-Co-TCPP, Au NCs + Co-TCPP, pristine Au NCs and Co-TCPP. Reprinted with permission from ref.¹²⁷. copyright 2022 Royal Society of Chemistry. **e** Light-driven catalytic durability over Au₂₅@Cu-BTC. Each cycle takes 8 h. Reprinted with permission from ref.¹²⁸. copyright 2023 Royal Society of Chemistry. **f** Schematic of the ‘covalence bridge’ catalysis strategy. Reprinted with permission from ref.¹²⁹. copyright 2022 Tshingua University.

Recently, such ultrasmall metal NCs with core size of ≤ 3 nm have been demonstrated to be a new type of effective antibacterial agent^{141,133}. Excellent photodynamic antibacterial efficacy of metal NC-based photocatalysts demands good photostability, outstanding harvesting ability for visible light, and superior productivity/separation of sufficient charge carriers. Interestingly, several types of AgNCs and AuNCs have been successfully designed as wide-spectrum antibacterial agents with improved bacterial killing effect. Such as, Wang et al. developed a AgNCs@CH-MF hybrid hydrogel (MF: mangiferin, CH – Chitosan) based on AgNCs and CH-MF, which shows significant synergistic and multiple impacts on bactericidal performance¹³⁰. Besides interfacial design of the hydrogel, the AgNCs@CH-MF hydrogel's strong antibacterial efficacy (fig 8a), was further improved due to the self-degradation into microparticles. AgNCs@CH-MF hybrid hydrogel possessed excellent biocompatibility by enhancing cell proliferation and prompting tissue regeneration. The AgNCs@CH-MF hydrogel has many advantages such as injectability, adequate swelling, good degradability, and good controlled release properties and shows antibacterial activity against both gram-positive and gram-negative bacteria by killing the adhered bacteria on the hydrogel surface with a high local concentration of Ag species. Similarly, Zheng et al. synthesized the MXene-AuNCs conjugated system which achieves excellent synergistic antimicrobial activity as MXenes pierce the bacterial membrane and conjugated AuNCs would be better internalized inside bacteria to generate reactive oxygen species (ROS) for disrupting bacterial normal metabolism¹³¹. Furthermore, the MXene nanosheets can also induce oxidative stress in bacteria, the resulting ROS are concentrated locally and function as a ROS reservoir, allowing for continual oxidation of bacterial membrane lipid for accelerated membrane breakdown and bacterial DNA for violent fragmentation, which ultimately results in the death of bacteria. In another work, Wang et al. have developed AgNCs@ELB (ELB: extract of Luria-Bertani (LB) medium) nanohybrid with

antibacterial characteristics through the encapsulation of AgNCs within the sacrificial ELB species using a facile light irradiation process. The Ag NCs@ELB upon being swallowed by the bacteria, could be effectively triggered once its ELB shell was digested by the bacteria. The as-designed AgNCs@ELB is highly efficient and biocompatible because of the synergistic effects of sacrificial ELB vehicle together with AgNCs¹³². Liu et al. demonstrated CDs@AuAg NCs (CD: carbon dot), exhibited enhanced photodynamic antibacterial performance (fig 8b), as conjugation of CD with AuAg NCs could enhance the harvest of visible light also ROS production and because of the small size and good water-solubility, CDs@AuAg NCs could promote their interaction with bacteria¹³³. The probable mechanism suggested by the authors is that upon visible-light illumination, the CDs' electrons might be excited, and some of them might then move from the conduction band (CB) level of the CDs to the LUMO level of the AuAg NCs through a charge transfer route, facilitating their surface photocatalytic interaction with dissolved oxygen (O₂) to produce ROS. Zhu et al., prepared TiO₂-NH₂@Au NC (fig 8c) nanohybrid by in which the chemically grafted AuNCs on the TiO₂-NH₂ surface can serve as a high-efficiency photosensitizer to harvest visible light to promote the charge carriers (e⁻/h⁺ pairs) generation. After that, the photoexcited e⁻ would transfer from the excited AuNCs to the CB of TiO₂ via amide bonds and finally migrate to the surface of TiO₂, reacting with the dissolved oxygen (O₂) for the production of ROS. In addition, the electron-rich amino groups on the TiO₂-NH₂ surface can further enhance their surface charge density via charge transfer eventually promoting the production of ROS, resulting in an excellent photodynamic antibacterial agent¹³⁴. The antibacterial activity can be explained by the highly luminescent AuNCs acting as a visible-light sensitizer for the photodynamic killing of bacteria, and the electrostatic interaction between TiO₂ and the AuNCs may make some of them as photocatalytic entities and facilitate the generation of ROS for killing the bacteria (fig 8d). Zheng et al. reported a unique magnetically oriented Ho-GO-Au nanohybrid system which utilizes both physical (via oriented GO) and chemical (via GO and Au NCs) mechanisms to perform as a bacterial killing agent¹³⁵. Under weak magnetic fields, the Ho-GO-Au nanosheets were able to be vertically orientated, providing high-density sharp edges with favoured orientation to successfully damage bacterial membranes. The conjugated AuNCs were efficiently delivered into GO-cut bacteria and induced high oxidative stress, which strongly disturbed bacterial metabolism, leading to the death of the bacteria.

Fig. 8 **a** Antibacterial performance of hydrogels against various bacteria. (I) The antibacterial activities of pristine Ag NCs (for reference, 0.17 mM), blank CH hydrogel (for reference), the CH-MF hydrogel and the AgNCs@CH hydrogel (with the Ag NC loading of 0.17 mM) against *S. aureus*. (II) Antibacterial activities and (III) corresponding antibacterial summary of blank CH hydrogel (as control) and AgNCs@CH-MF against *S. aureus*, *B. subtilis*, *E. coli*, and *P. aeruginosa* based on LB-agar plate counting. Reprinted with permission from ref.¹³⁰. copyright 2021 Elsevier. **b** Bacterial colony growth of (I) *S. aureus* and (II) *E. coli* in the presence of AuAg NCs, CDs, CDs@AuAg NCs, and water (as control) under visible-light illumination or dark conditions for 40 min. Bactericidal efficiencies of (III) Gram-positive *S. aureus* and (IV) Gram negative *E. coli* treated with AuAg NCs, CDs, CDs@AuAg NCs, and water (as control) under visible-light illumination or dark conditions. Reprinted with permission from ref.¹³³. copyright 2022 Royal Society of Chemistry. **c** (I and II) TEM images of the TiO₂-NH₂@Au NC antibacterial agent with different magnifications. (III) EDS mapping image of the TiO₂-NH₂@Au NCs and the corresponding elemental mappings of Ti, O, Au, N and S elements. **d** The proposed mechanism of the visible-light-driven antibacterial process of TiO₂-NH₂@Au NCs. Reprinted with permission from ref.¹³⁴. copyright 2021 Royal Society of Chemistry.

Comment 3.

The authors stated that “but very few have been characterized by single x-ray crystallography, such as ...”, but now many hydrophobic thiolated ligands-protected Au nanoclusters have been characterized

by this technique. Therefore, a suitable revision in the expression is expected to avoid possible confusion or misunderstanding.

Response: Thank you for pointing this out. We revised the sentence as follow:

Numerous numbers of NCs are now discovered with their detailed mass spectrum, and many of them have been characterized by single x-ray crystallography, such as $\text{Au}_{18}(\text{SR})_{14}$, $\text{Au}_{20}(\text{SR})_{16}$, $\text{Au}_{23}(\text{SR})_{16}$, $\text{Au}_{24}(\text{SR})_{16}$, $\text{Au}_{25}(\text{SR})_{18}$, $\text{Au}_{28}(\text{SR})_{20}$, $\text{Au}_{30}(\text{SR})_{18}$, etc.^{1, 3}

Comment 4.

Several typos, and format issues are observed in the context and Reference section.

Response: Thank you for pointing this out. We have checked and corrected this issue in the revised MS.

Reviewer #2

Comment 1:

The Communication is a very well edited summary article. It summarises the background to the topic well. My small comment is that some of the figures are quite small, the captions are not visible, which makes it difficult for the reader to understand the data. please enlarge the size of the very small letters.

Response: We thank the reviewer for this suggestion and corresponding Figures and text sizes are changed for better visualization in the revised MS.

Fig.2 a TEM image of the crossed assembly formed with Ag₄₄@Te NWs for the same NW concentration as the pristine Te NWs. and (I) a sandwich structure of the nanohybrid, (II) H-bonding between neighboring clusters in the Ag₄₄(p-MBA)₃₀ superlattice structure and (III) Schematic representation of the 81° orthogonal bilayer

assembly, respectively. Reprinted with permission from ref.⁸¹. Copyright 2016 WILEY. **b** TEM images of the AgNC@ZnO Tp hybrid at different magnifications (I-III). **c** Confocal laser scanning microscopy (CLSM) images of the AgNC@ZnO Tp excited at 3.06 eV/405 nm; (I) showing a PL below/above 2.0 eV/615 nm, (II) bright-field image, and (III) the overlay of bright-field and PL intensity images. Reprinted with permission from ref.⁸². Copyright 2021 The Royal Society of Chemistry. **d** Schematic illustration of the synthesis processes for (a) Au₂₅(SG)₁₈@ZIF-8 (assembling of Au₂₅(SG)₁₈ into ZIF-8 matrix) and (b) Au₂₅(SG)₁₈/ZIF-8 (impregnating Au₂₅(SG)₁₈ on the outer surface of ZIF-8) HAADF-STEM images of (I,II) 1000 Au₂₅(SG)₁₈@ZIF-8, and (I',II') 1000 Au₂₅(SG)₁₈/ZIF-8. Reprinted with permission from ref.⁸³. Copyright 2017 WILEY.

Fig. 3 a (I) Representation of the self-assembly of Ag₄₄ (curved arrow directs to the ESI MS of the cluster) on

the GNR@pMBA surface, (II) TEM image of GNR@pMBA (inset shows an HRTEM image of a single nanorod), (III) TEM image of GNR@Ag₄₄ (inset shows dark field STEM image of a composite particle) **b** TEM 3D reconstructed structure of GNR@Ag₄₄, showing octahedral geometry. Reprinted with permission from ref.⁸⁴. Copyright 2018 Wiley. **c** Interaction between AuNT@pMBA and Ag₄₄: (I) Schematic representation of the H-bond mediated assembly of Ag₄₄ on AuNT@pMBA template, (II) TEM image of AuNT@pMBA@Ag₄₄ after 24 h, (III) that of a single AuNT@pMBA@Ag₄₄ clearly showing core-shell composite structure. Reprinted with permission from ref.⁸⁵. Copyright 2023 Royal Society of Chemistry.

Fig: 4 **a** (I) HAADF-STEM image of the Au₁₂Ag₃₂(SR)₃₀@ZIF-8 composite; (II) TEM images of the Au₁₂Ag₃₂(SR)₃₀@ZIF-8 composite, (III) the Au₂₄Ag₄₆(SR)₃₂@ZIF-8 composite, (IV) the Ag₄₄(SR)₃₀@ZIF-8 composite, and (V) the Ag₁₂Cu₂₈(SR)₃₀@ZIF-8 composite. Reprinted with permission from ref.⁸⁶. copyright 2018 Royal Society of Chemistry. **b** single AuNR@mSiO₂@Ag₂₉, and a 3D reconstructed image of single AuNR@mSiO₂@Ag₂₉, where some of the atoms from the front part are omitted to give a clear core-shell view. Reprinted with permission from ref.⁸⁷. copyright 2022 American Chemical Society. **c** Illustration showing the luminescent Cu NCs-ImRGO nanocomposite. Reprinted with permission from ref.⁸⁸. copyright 2017 American Chemical Society. **d** Schematic design of hierarchically structured Au_n NCs catalysts Au_n/Ni_xMg_{3-x}Al-T (taking Au₂₅/Ni_{1.5}Mg_{1.5}Al-280 as an example). Reprinted with permission from ref.⁸⁹. copyright 2022 Elsevier Inc. **e** (I) Schematic representation of the formation of the organo-inorganic CCH material. The inset shows the TEM image (i) and photograph (ii) of the intense red luminescent aggregates formed during the reaction of the clay and cluster. The schematic of the aminoclay sheet is presented for representation purposes only. (II) TEM image of the aminoclay sheet. (III) MALDI MS of the Au₃₀@BSA. (IV) Optical (left) and fluorescence (right) images of CCH. (V) Luminescence spectrum of CCH showed a maximum at 650 nm for 365 nm excitation. Reprinted with permission from ref.⁹⁰. copyright 2021 American Chemical Society.

Fig. 5 Effect of MALDI TOF mass spectra of $\text{Au}_{25}\text{SBB}_{18}$ (black trace) with increasing SBB/CD ratio (green to brown trace) in solution. Schematic cluster representations with different amounts of CD inclusions are also shown. At 1:0.05, some parent $\text{Au}_{25}\text{SBB}_{18}$ is also seen, shown with #. UV-vis absorption spectra **b** and luminescence spectra ($\lambda_{\text{ex}} = 992 \text{ nm}$) **c** of the $\text{Au}_{25}\text{SBB}_{18}$ cluster with increasing amounts of CD inclusion. Reprinted with permission from ref.⁹¹. copyright 2013 American Chemical Society. **d** Schematic representation of “ideal” $\text{Au}_{133}(\text{SR})_{52}$ organization in the periphery of gradient bMOF-102/106. **e** Absorbance spectra of bMOF-102/106 before (black) and after (red) encapsulation of $\text{Au}_{133}(\text{SR})_{52}$ and corresponding optical images (before, top; after, bottom). **f** Absorbance spectra of different regions of a bMOF-102/106 crystal showing the presence of $\text{Au}_{133}(\text{SR})_{52}$ in the periphery (orange) but not the core (blue) and corresponding optical images (periphery, top; core, bottom). Reprinted with permission from ref.⁹². copyright 2016 American Chemical Society. **g** Schematic representation of entrapment of atomically precise clusters in cyclodextrin-functionalized aminoclay (AC-CD). Reprinted with permission from ref.⁹³. copyright 2020 American Chemical Society. **h** Drift time profile of ($n=2$) complexes and its corresponding CCS value indicates the cis-trans isomers of $n=2$. Reprinted with permission from ref.⁹⁴. copyright 2018 American Chemical Society. **i** CID study on $[\text{X}(\text{C}_{60})]^{3-}$ at increasing collision energies (CE in instrumental units) of (I), (II), and (III). **j** CID spectrum of $[\text{X}(\text{C}_{60})_n]^{3-}$ ($n = 2-4$) are shown in (I-III) respectively, at CE 10, $\text{X} = \text{Ag}_{29}(\text{BDT})_{12}$. Reprinted with permission from ref.⁹⁵. copyright 2018 American Chemical Society.

Fig. 6 pH-Dependent exfoliation and hybridized functionalization of MoS₂ nanosheets with BSA-caged Au₂₅ clusters (Au₂₅-BSA). **a** UV-vis absorption spectra of MoS₂ nanosheets exfoliated at different pH values with BSA as an effective cleaving agent. **b** The evolution of absorption intensity of MoS₂ nanosheets as a function of pH together with the corresponding optical images. **c** Zeta potentials of MoS₂ and pure BSA in different pH environments. **d** Schematic exfoliation of MoS₂ nanosheets and subsequent surface growth of Au₂₅ clusters into Au_m nanoparticles. **e** Low-resolution and **f** High-resolution TEM images of Au_m/MoS₂ nanosheets, and **g** enlarged TEM image of the highlighted area in **f**. Reprinted with permission from ref.⁹⁶. copyright 2018 Royal Society of Chemistry.

Comment 2.

A short thought on the quantum efficiency and lifetime data of fluorescent noble metal nanoclusters at the beginning of the article would be nice. After these minor corrections, I recommend the publication of this nice work.

Response: We thank the reviser for this suggestion. A brief description of the fluorescence properties of these nanoclusters is highlighted in the introduction section of the revised MS.

The photoluminescence quantum yields (PLQY) of metal nanoclusters are relatively low in comparison to corresponding semiconductor quantum dots (QDs), and their emission lifetime is typically in the nanoseconds (ns) regime²². However, these materials exhibit good photo-stability, biocompatibility, facile synthesis techniques, large Stokes shifts, and tunable fluorescence intensity^{6, 34, 35, 36, 37, 38, 39, 40, 41}. There are now many reported methods to improve their PLQY, especially via doping, surface rigidification⁴² or ligand engineering¹⁶ which makes them useful for many PL-based applications^{1, 43}.

Reviewer #3

Comment 1.

This work summarized the synthesis of nanohybrid based on the interaction between NCs and supports, but not mentioned the properties and application. The potential application of nanohybrid is more important to guild the oriented synthesis. The proportion of forward-looking and/or speculative is less than the review in the text.

Response: We thank the reviewer for this suggestion, which helped us improve the quality of our review. We have added a separate section for applications in the revised manuscript.

Applications of Nanohybrids:

These nanohybrids exhibit synergistic/anti-synergistic properties and even some new properties depending on the type of interactions between the NCs and the other nanomaterials. This opens up the possibility to use such nanohybrids in a broad variety of novel applications starting from energy to bio-applications. This review will briefly summarize the potential uses of NCs-based nanohybrids in some emerging fields such as photovoltaics^{121, 122, 123, 124}, photocatalysis^{125, 126, 127, 128, 129}, antimicrobials^{130, 131, 132, 133, 134, 135}.

The interesting properties such as size dependent emission spectra, distinct absorption features, non-toxic behaviour, scalable and ambient synthesis procedures of metal NCs makes them potentially material for photovoltaic applications, such as light-emitting diodes (LEDs). However, those NCs suffer from instability, low photoluminescence intensity, and a single distinguishable emission colour, which also makes it challenging to use for such applications. However, nanohybrids consisting of metal NCs and other functional nanomaterials can address those challenges and can be used for high-performance LED applications. For example, Yin et al. reported a high-performance white light emitting diode (WLED) based on AuAg bimetallic NCs with a high CRI (80.6), high LE (91.9 lm/W), and nonchromaticity parameter drift¹²¹. The bare AuAg bimetallic NCs have a bright orange emission (610nm), large Stokes shift, and a high PLQY (43% in solid-state), making them appropriate for high-end lighting applications. When orange emitting AuAg bimetallic NCs and green $Y_3(Al, Ga)_5O_{12}:Ce^{3+}$ phosphors (Ga-YAG phosphor) (emission maxima 530nm) was combined with a molar ratio of 1:10 and integrated the mixture on a blue-emitting (emission maximum 455 nm) InGaN-based LED chip, the resulting hybrid material was able to function as a trichromatic white light-emitting diode (WLED) (Fig. 7a). Here the orange light emitted from the AuAg nanocrystals, green light from the Ga-YAG phosphor, and blue light from the InGaN chip were combined to produce white light. While, the correlated colour temperature (CCT) of white light-emitting diodes (WLEDs) can be tailored to a desired range by adjusting the proportions of nanocrystals and Ga-YAG, producing warm light and more effective for human eyes and many other creatures. Bhandari et al. also prepared bio-friendly luminescent white light emitting NC-based nanohybrid composed of red emitting Au NCs, green emitting ZnQ₂ complex (a complex of Mn²⁺ doped ZnS Qdot and 8- hydroxyquinoline (HQ)) and blue emitting protein¹²². Wang et al. have successfully synthesized biocompatible HGC/CuNCs films with exceptional luminescence properties¹²³. The fabrication involved utilizing negatively charged CuNCs as a component of a positively charged biocompatible graft copolymer, specifically hypromellose grafted chitosan (HGC) The HGC with a positive charge served a dual purpose in inducing the aggregation of CuNCs and serving as the matrix for developing films with exceptional luminescent properties (Fig. 7b). Compared to bare CuNCs, the resultant film exhibits an increase in PLQY from 0.5% to 42%, an improvement in PL average time from 6.1 μs to 25.9 μs, and a blue shift in the PL spectrum (649nm to 600nm). These films are used as color converters to fabricate down conversion

remote LED. Yang et al. developed fluorescent GSH-CuNCs/Zn-HDS (Hydroxy double salts) nanohybrid materials using surface CIEE (Confinement induced enhanced emission) based on Zn-HDS as host material and GSH-CuNCs as guest molecules¹²⁴. The resulting hybrid exhibit high PLQY and longer fluorescence life time and higher stability than the bare CuNCs for LED applications. GSH-CuNCs/Zn-HDS powder was coated on the commercial UV LED chip provides orange emitting LED with luminance efficiency (1.5), color purity (85%), CCT (2461). The performance of this LED device against varying driving currents illustrates its potential for use in real-world application.

Atomically precise metal NCs with ultrasmall size emerge as efficient photocatalysts by virtue of their unique electronic and optical properties, high surface-to-volume ratio, and abundance of active sites^{79, 129}. Still, the photocatalytic performance has been affected by several limitations like light-driven aggregation of metal NCs, Lack of active sites for catalysis, and quick recombination of the photogenerated electrons and holes^{127, 128}. Many studies have been done to improve photocatalytic efficiency by using NC-based hybrid materials. For example, Negishi et al. demonstrated that hybridization of BaLa₄Ti₄O₁₅ with tiny (1.2±0.3 nm) Au₂₅(SG)₁₈ NCs results in 2.6 times higher water splitting activity than hybridization of BaLa₄Ti₄O₁₅ with larger Au nanoparticles (10-30 nm), which can be attributed to the distinct surface and electronic effects of atomically precise gold NCs. The gold NC acts as a co-catalyst and provides a more significant number of active sites for catalysis even with lower loading, resulting in increased catalytic activity. Zhu et al. designed a hybrid photocatalyst-Pt@cyclodextrin NCs on C₃N₄/MXene (Ti₃C₂) heterojunction (Pt@b-CD/C₃N₄-M) which delivers high H₂O₂ production of 147.1 μM L⁻¹ (~6 times greater H₂O₂ production with contrast to pristine C₃N₄)¹²⁶. This excellent improvement in catalytic activity clearly reflects the complementary contribution of Pt cores where Pt acts as an active sites for catalysis and photogenerated e⁻ acceptor, per-6-thio-β-cyclodextrin acts as hydrophobic delivery channel for O₂ transfer, C₃N₄ as photoinduced electron generator, Mxene as improved visible light absorber (Fig. 7c). Similarly, Xue et al. reported a covalently hybridized Au-Co-TCPP hybrid photocatalyst which provides 2.2 times higher H₂O₂ production capability than the pure Au NCs by leveraging the synergistic contribution of the hybrid structure (Fig. 7d). Au-Co-TCPP exhibits a H₂O₂ production of 235.93 mM in 60 min, whereas the pristine Au NCs produce only 107.11 mM of H₂O₂¹²⁷. Here, in addition to improving visible light absorption of AuNCs, the grafted Co-TCPP unit may operate as electron acceptors, thereby effectively enhancing the charge separation of Au NCs while also offering an abundant supply of active sites for photocatalysis. As the recyclability or reusability of a catalyst is directly connected to the stability of the catalyst, encapsulation of metal NC or hybridization with a supporting framework is a very effective way to enhance the stability of the photocatalyst as well as addressing the issue of recyclability. So, Zhang et al. designed a photocatalyst by encapsulating Au₂₅(p-MBA)₁₈ (p-MBA = 4-mercaptobenzoic acid) in a Cu₃(BTC)₂ (BTC = benzene-1,3,5-tricarboxylate) metal-organic framework (Au₂₅@Cu-BTC) which prevents the aggregation of AuNCs¹²⁸. The catalyst exhibits steady performance for successive 6 cycles (48 h), representing its outstanding stability (Fig. 7e). Besides, “covalence bridge” between the two components of the hybrid NC is an effective strategy to boost both the activity and durability (Fig. 7f)¹²⁹.

Fig. 7 **a** Schematic illustration of a representative WLED constructed by combining AIE-Featured orange-emitting AuAg NCs with green phosphors on a blue chip. Reprinted with permission from ref.¹²¹. copyright 2020 American Chemical Society. **b** (I) Schematics of a conventional (left) and a remote type (right) down-conversion LED; (II) Emission spectrum of a remote LED employing the HGC/Cu NC film; inset on the left provides CRI, CCT, and CIE chromaticity coordinate of the device, and inset on the right shows a photograph of the working orange emitting device. Reprinted with permission from ref.¹²³. copyright 2018 WILEY-VCH. **c** proposed mechanism for photocatalytic H₂O₂ production over the Pt@b-CD/C₃N₄-M photocatalyst. Reprinted with permission from ref.¹²⁶. copyright 2021 Royal Society of Chemistry. **d** Time-course photocatalytic H₂O₂ production over Au-Co-TCPP, Au NCs + Co-TCPP, pristine Au NCs and Co-TCPP. Reprinted with permission from ref.¹²⁷. copyright 2022 Royal Society of Chemistry. **e** Light-driven catalytic durability over Au₂₅@Cu-BTC. Each cycle takes 8 h. Reprinted with permission from ref.¹²⁸. copyright 2023 Royal Society of Chemistry. **f** Schematic of the ‘covalence bridge’ catalysis strategy. Reprinted with permission from ref.¹²⁹. copyright 2022 Tshingua University.

Recently, such ultrasmall metal NCs with core size of ≤ 3 nm have been demonstrated to be a new type of effective antibacterial agent^{141,133}. Excellent photodynamic antibacterial efficacy of metal NC-based photocatalysts demands good photostability, outstanding harvesting ability for visible light, and superior productivity/separation of sufficient charge carriers. Interestingly, several types of AgNCs and AuNCs have been successfully designed as wide-spectrum antibacterial agents with improved bacterial killing effect. Such as, Wang et al. developed a AgNCs@CH-MF hybrid hydrogel (MF: mangiferin, CH – Chitosan) based on AgNCs and CH-MF, which shows significant synergistic and multiple impacts on bactericidal performance¹³⁰. Besides interfacial design of the hydrogel, the AgNCs@CH-MF hydrogel's strong antibacterial efficacy (fig 8a), was further improved due to the self-degradation into microparticles. AgNCs@CH-MF hybrid hydrogel possessed excellent biocompatibility by enhancing cell proliferation and prompting tissue regeneration. The AgNCs@CH-MF hydrogel has many advantages such as injectability, adequate swelling, good degradability, and good controlled release properties and shows antibacterial activity against both gram-positive and gram-negative bacteria by killing the adhered bacteria on the hydrogel surface with a high local concentration of Ag species. Similarly, Zheng et al. synthesized the MXene-AuNCs conjugated system which achieves excellent synergistic antimicrobial activity as MXenes pierce the bacterial membrane and conjugated AuNCs would be better internalized inside bacteria to generate reactive oxygen species (ROS) for disrupting bacterial normal metabolism¹³¹. Furthermore, the MXene nanosheets can also induce oxidative stress in bacteria, the resulting ROS are concentrated locally and function as a ROS reservoir, allowing for continual oxidation of bacterial membrane lipid for accelerated membrane breakdown and bacterial DNA for violent fragmentation, which ultimately results in the death of bacteria. In another work, Wang et al. have developed AgNCs@ELB (ELB: extract of Luria-Bertani (LB) medium) nanohybrid with

antibacterial characteristics through the encapsulation of AgNCs within the sacrificial ELB species using a facile light irradiation process. The Ag NCs@ELB upon being swallowed by the bacteria, could be effectively triggered once its ELB shell was digested by the bacteria. The as-designed AgNCs@ELB is highly efficient and biocompatible because of the synergistic effects of sacrificial ELB vehicle together with AgNCs¹³². Liu et al. demonstrated CDs@AuAg NCs (CD: carbon dot), exhibited enhanced photodynamic antibacterial performance (fig 8b), as conjugation of CD with AuAg NCs could enhance the harvest of visible light also ROS production and because of the small size and good water-solubility, CDs@AuAg NCs could promote their interaction with bacteria¹³³. The probable mechanism suggested by the authors is that upon visible-light illumination, the CDs' electrons might be excited, and some of them might then move from the conduction band (CB) level of the CDs to the LUMO level of the AuAg NCs through a charge transfer route, facilitating their surface photocatalytic interaction with dissolved oxygen (O₂) to produce ROS. Zhu et al., prepared TiO₂-NH₂@Au NC (fig 8c) nanohybrid by in which the chemically grafted AuNCs on the TiO₂-NH₂ surface can serve as a high-efficiency photosensitizer to harvest visible light to promote the charge carriers (e⁻/h⁺ pairs) generation. After that, the photoexcited e⁻ would transfer from the excited AuNCs to the CB of TiO₂ via amide bonds and finally migrate to the surface of TiO₂, reacting with the dissolved oxygen (O₂) for the production of ROS. In addition, the electron-rich amino groups on the TiO₂-NH₂ surface can further enhance their surface charge density via charge transfer eventually promoting the production of ROS, resulting in an excellent photodynamic antibacterial agent¹³⁴. The antibacterial activity can be explained by the highly luminescent AuNCs acting as a visible-light sensitizer for the photodynamic killing of bacteria, and the electrostatic interaction between TiO₂ and the AuNCs may make some of them as photocatalytic entities and facilitate the generation of ROS for killing the bacteria (fig 8d). Zheng et al. reported a unique magnetically oriented Ho-GO-Au nanohybrid system which utilizes both physical (via oriented GO) and chemical (via GO and Au NCs) mechanisms to perform as a bacterial killing agent¹³⁵. Under weak magnetic fields, the Ho-GO-Au nanosheets were able to be vertically orientated, providing high-density sharp edges with favoured orientation to successfully damage bacterial membranes. The conjugated AuNCs were efficiently delivered into GO-cut bacteria and induced high oxidative stress, which strongly disturbed bacterial metabolism, leading to the death of the bacteria.

Fig. 8 **a** Antibacterial performance of hydrogels against various bacteria. (I) The antibacterial activities of pristine Ag NCs (for reference, 0.17 mM), blank CH hydrogel (for reference), the CH-MF hydrogel and the AgNCs@CH hydrogel (with the Ag NC loading of 0.17 mM) against *S. aureus*, (II) Antibacterial activities and (III) corresponding antibacterial summary of blank CH hydrogel (as control) and AgNCs@CH-MF against *S. aureus*, *B. subtilis*, *E. coli*, and *P. aeruginosa* based on LB-agar plate counting. Reprinted with permission from ref.¹³⁰ copyright 2021 Elsevier. **b** Bacterial colony growth of (I) *S. aureus* and (II) *E. coli* in the presence of AuAg NCs, CDs, CDs@AuAg NCs, and water (as control) under visible-light illumination or dark conditions for 40 min. Bactericidal efficiencies of (III) Gram-positive *S. aureus* and (IV) Gram negative *E. coli* treated with AuAg NCs, CDs, CDs@AuAg NCs, and water (as control) under visible-light illumination or dark conditions. Reprinted with permission from ref.¹³³ copyright 2022 Royal Society of Chemistry. **c** (I and II) TEM images of the TiO₂-NH₂@Au NC antibacterial agent with different magnifications. (III) EDS mapping image of the TiO₂-NH₂@Au NCs and the corresponding elemental mappings of Ti, O, Au, N and S elements. **d** The proposed mechanism of the visible-light-driven antibacterial process of TiO₂-NH₂@Au NCs. Reprinted with permission from ref.¹³⁴ copyright 2021 Royal Society of Chemistry.

Comment 2.

The category of nanohybrid is kind of unreason. It can be classified by the type of interaction or the type of support et al. The interaction could be covalence bond and non-covalence bond (electrostatic

interaction, pi-pi interaction, hydrogen bond, and other Van der Waals intermolecular force et al.) In the text, the host-guest interaction is a big scope including both the covalence and non-covalence interaction, while the hydrophobicity is more a property but not an interaction force.

Response: Thank you for your precise comments, which assisted us in making our review work more thorough. Based on the reviewer's suggestion, we have modified the category of nanohybrids and revised the text accordingly. Few introductory lines about the classifications are also added in the revised manuscript.

In general, hybrid nanomaterials are made up of two or more organic or inorganic components that are typically linked at the nanoscale level by covalent bonds or noncovalent bonds (electrostatic interactions, H-bonding, van der Waals interactions, and so on)⁸⁰. Integrating diverse materials within a single material yields hybrid materials with multifunctional and distinctive characteristics. This is attributed to the emergence of new or synergistic/anti-synergistic effects in their properties, which are not present in the individual components. Different strategies are reported to synthesize NCs-based nanohybrids, which can be categorized based on the type of interactions between the NCs and other nanomaterials, e.g., class-I nanohybrids (covalent interactions), class-II nanohybrids (noncovalent interactions such as H-bonding interactions, electrostatic interactions), and class-III nanohybrids (other types of interactions such as hydrophobic interactions, host-guest interactions, etc.)

Comment 3.

The author name of ref. 88 in the text is inconsistent with the reference. Therefore, some revision is needed, and it is not suitable for publication as it is.

Response: We thank the reviewer for his encouraging comments to improve the quality of the review. The references (including ref. no. 88 in the older version) and the formats were revised, and we hope this revised version will be suitable for publication.

REVIEWERS' COMMENTS:

Reviewer #1 (Remarks to the Author):

The authors have carefully revised the manuscript according to reviewers' comments/suggestions, and currently it can be accepted by the journal.

Reviewer #3 (Remarks to the Author):

I think the points raised in my previous comments have been addressed in the revised MS so that I recommend its publication as is.